# WASP-mediated regulation of anti-inflammatory macrophages is IL-10 dependent and is critical for intestinal homeostasis

Amlan Biswas[1,2,3], Dror S. Shouval[1,3,4,5], Alexandra Griffith[1,3], Jeremy A. Goettel[1,2,3], Michael Field [1,3], Yu Hui Kang [1,2,3], Liza Konnikova[1,2,3], Erin Janssen[2,6], Naresh Singh Redhu[1,2,3], Adrian J. Thrasher[7], Talal Chatila [2,6], Vijay K. Kuchroo[8], Raif S Geha[2,6], Luigi D. Notarangelo[9], Sung-Yun Pai[10], Bruce H. Horwitz[2,3,,11,12] & Scott B. Snapper[1,3,13]

Mutations in Wiskott–Aldrich syndrome protein (WASP) cause autoimmune sequelae including colitis. Yet, how WASP mediates mucosal homeostasis is not fully understood. Here we show that WASP-mediated regulation of anti-inflammatory macrophages is critical for mucosal homeostasis and immune tolerance. The generation and function of anti-inflammatory macrophages are defective in both human and mice in the absence of WASP. Expression of WASP specifically in macrophages, but not in dendritic cells, is critical for regulation of colitis development. Importantly, transfer of WT anti-inflammatory macrophages prevents the development of colitis. DOCK8-deficient macrophages phenocopy the altered macrophage properties associated with WASP deficiency. Mechanistically, we show that both WASP and DOCK8 regulates macrophage function by modulating IL-10-dependent STAT3 phosphorylation. Overall, our study indicates that anti-inflammatory macrophage function and mucosal immune tolerance require both WASP and DOCK8, and that IL-10 signalling modulates a WASP-DOCK8 complex.

[1] Division of Gastroenterology, Hepatology and Nutrition, Boston Children's Hospital, 300 Longwood Avenue, Boston, Massachusetts 02115, USA. [2] Department of Pediatrics, Harvard Medical School, 25 Shattuck Street, Boston, Massachusetts 02115, USA. [3] VEO-IBD Consortium, 300 Longwood Avenue, Boston, MA 02115, USA. [4] Division of Pediatric Gastroenterology and Nutrition, Edmond and Lily Safra Children's Hospital, Sheba Medical Center, Tel Hashomer 52621, Israel. [5] Sackler Faculty of Medicine, Tel Aviv University, Tel Aviv 6997801, Israel. [6] Division of Immunology, Boston Children's Hospital, Boston, 1 Blackfan Circle, Massachusetts 02115, USA. [7] Great Ormond Street Hospital NHS Trust, London and Institute of Child Health, University College London, 30 Guilford Street, London WC1N 1EH, UK. [8] Evergrande Center for Immunologic Diseases, Harvard Medical School and Brigham and Women's Hospital, 60 Fenwood Road, Boston, Massachusetts 02115, USA. [9] Clinical Immunology and Microbiology, NIAID, National Institutes of Health, 10 Center Drive, MSC 1456, Bethesda, Maryland 20892-9806, USA. [10] Division of Hematology-Oncology, Boston Children's Hospital Boston, 1 Blackfan Circle, Boston, Massachusetts 02115, USA. [11] Department of Pathology, Brigham and Women's Hospital, 75 Francis Street, Boston 02115 Massachusetts, USA. [12] Division of Emergency Medicine, Boston Children's Hospital, Boston, 300 Longwood Avenue, Boston, Massacusetts 02115, USA. [13] Division of Gastroenterology, Brigham and Women's Hospital, Department of Medicine, Harvard Medical School, 75 Francis Street, Boston, Massachusetts 02115, USA. Correspondence and requests for materials should be addressed to S.B.S. (email: scott.snapper@childrens.harvard.edu)

A large genome-wide association study among inflammatory bowel disease (IBD) patients identified over 163 loci associated with IBD risk[1]. A Bayesian network analysis containing these risk loci as well as gene expression data identified an IBD sub-network that includes several genes (e.g., *IL10*, *NOD2*, *HCK* and *WAS*) that are highly enriched in bone-marrow-derived macrophages (BMDMs) and point to a possible role for this IBD sub-network in regulating anti-inflammatory macrophage development and/or function[1]. The Wiskott–Aldrich syndrome (*WAS*) gene, encoding the actin cytoskeletal protein WAS protein (WASP), is one of the genes identified within this sub-network along with other IBD-associated genes. Patients with WAS typically manifest recurrent infections, thrombocytopenia and eczema. In addition, 10% of patients develop IBD and 100% of $Was^{-/-}$ mice on the 129SvEv background develop spontaneous colitis[2–4]. WASP expression is restricted to haematopoietic lineages and broad defects are observed in most WASP-deficient leukocytes[5]. WASP regulates cytoskeleton-dependent functions, including podosome formation, migration, phagocytosis and antigen uptake in a variety of innate immune cells[6–11]. Our group has previously reported that $Was^{-/-}$ innate immune cells are a primary driver of intestinal inflammation[12]. $Was^{-/-}Rag2^{-/-}$ mice rapidly lose weight and develop severe colitis after transfer of unfractionated WT CD4[+] T cells, whereas $Rag2^{-/-}$ mice that express WASP do not develop colitis[12]. Together, these studies suggest that WASP function within an innate immune cell is necessary to avert intestinal inflammation. However, the precise identity of the innate immune population that requires WASP to prevent inflammation and the function of WASP within those cells, have not been previously determined.

Over the past two decades, our understanding of the diversity and unique nature of intestinal innate immune cells has been amplified considerably. Tissue resident innate immune cells including dendritic cells (DCs) and macrophages regulate immune responses directed toward mucosal microbes and other luminal antigens. CD103[+] CD11c[+] DCs facilitate immune tolerance by promoting FOXP3[+] regulatory T (Treg) cell differentiation and the production of retinoic acid and transforming growth factor (TGF)-β[13,14]. In addition, lamina propria (LP) CX3CR1[high]CD11b[+] CD11c[+] cells are a subset of regulatory myeloid cells, which suppress CD4[+] T-cell proliferation in a cell contact-dependent manner[15]. Several macrophage subsets have been identified and characterized that are distinct from classically activated macrophages[16]. In response to a variety of stimuli, these alternatively activated macrophages exhibit immunoregulatory function and produce high levels of the anti-inflammatory cytokine interleukin (IL)-10 with undetectable levels of the pro-inflammatory cytokine IL-12[16–18]. The immune-regulatory potential of these macrophages has been demonstrated in animal models of endotoxic shock, multiple sclerosis and IBD[18–20].

Here we show that WASP expression in macrophages is critical for the maintenance of intestinal immune tolerance and protection from colitis. $Was^{-/-}$ macrophages lose their tolerogenic properties and acquire a pro-inflammatory signature. Macrophage-specific deletion of WASP causes severe colitis in a naive CD4[+] T-cell transfer model. Importantly, we demonstrate that the generation and function of bone-marrow-derived anti-inflammatory macrophages require WASP. Similarly, patients with WAS exhibit impaired development and function of anti-inflammatory macrophages. Mechanistically, we show that IL-10 modulates a WASP:DOCK8-signalling complex. Collectively, these data demonstrate that WASP regulates intestinal homeostasis through modulation of anti-inflammatory macrophages.

## Results

**WASP regulates macrophage function and differentiation.** We sought to investigate the role of WASP in macrophages differentiation in both mucosal and non-mucosal sites. In the LP, monocytes undergo several stages of development during differentiation and can be categorized into four different groups based on the expression of Ly6c and major histocompatibility complex (MHC) II: P1 (Ly6c[hi] MHCII[−]), P2 (Ly6c[int to hi] MHC II[+]) and P3+ P4 (Ly6c[low] MHC II[+], P4 CX3CR1[+])[21] (Supplementary Fig. 1a). P2 LP macrophages have pro-inflammatory characteristics, whereas P3 and P4 LP macrophages have anti-inflammatory properties. To examine whether WASP regulates LP macrophage differentiation and function, and to minimize any effect that inflammation may have on skewing of macrophage differentiation, we compared the phenotype of colonic macrophages from pre-colitic 5-week-old $Was^{-/-}$ and wild-type (WT) mice. In these $Was^{-/-}$ mice we observed a significant increase in the percentage of P2 pro-inflammatory macrophages (***$p <$ 0.001, Student's $t$-test) and a concomitant decrease in the percentage of P3, P4 anti-inflammatory macrophages (***$p < 0.001$, Student's $t$-test) compared with WT mice (Fig. 1a). These alterations were more pronounced in 12-week-old $Was^{-/-}$ mice (Fig. 1b). Although the frequency of P2 versus P3/P4 macrophages was inversed in $Was^{-/-}$ mice compared with WT animals, the absolute number of all macrophages subset was greater in $Was^{-/-}$ mice compared with control animals, which is most likely due to increased recruitment of circulating monocytes in the setting of inflammation. Altered macrophage populations were also apparent in $Was^{-/-}Rag^{-/-}$ mice (129 SvEv background), which in the absence of T-cell transfer do not develop colonic inflammation (Supplementary Fig. 1b)[12]. To further characterize functionally LP macrophages, we evaluated the expression of pro- and anti-inflammatory genes expression in sorted P3 + P4 macrophages isolated from 5-week-old $Was^{-/-}$ and WT mice. The expression of anti-inflammatory genes including *Arg1*, *Fizz1*, *Ym1* and *Il10* was lower in $Was^{-/-}$ P3 + P4 macrophages compared with WT macrophages (Fig. 1c). Interestingly P3 + P4 macrophages from $Was^{-/-}$ mice was more pro-inflammatory in nature as evidenced by higher expression of inflammatory genes including *Il1b*, *Il23*, *Tnf* and *Il12* (Fig. 1c). The per cell expression of inflammatory genes in P2 macrophages were comparable between WT and $Was^{-/-}$ mice (Supplementary Fig. 1c).

To determine whether the aberrant differentiation of LP macrophages observed in $Was^{-/-}$ mice was due to a cell-intrinsic defect caused by the absence of WASP, we used a mixed chimera approach to study macrophages differentiation. Bone marrow from WT CD45.1[+] mice and $Was^{-/-}$ CD45.2[+] mice were transferred at a 1:1 ratio into irradiated CD45.2[+] $Was^{-/-}$ mice and analysed 8 weeks after reconstitution. Flow cytometric analysis of gated WT (CD45.1) and $Was^{-/-}$ (CD45.2) cells from LP showed an increase in the percentage of pro-inflammatory macrophages and a concomitant decrease in anti-inflammatory macrophages in $Was^{-/-}$ compared with WT compartments (Fig. 1d). Moreover, the P3/P4 macrophages from $Was^{-/-}$ (CD45.2) compartment were more pro- and less anti-inflammatory in nature compared with WT (CD45.1) compartment (Fig. 1e). Taken together, these results indicate that WASP regulates the development of LP anti-inflammatory macrophages in a cell intrinsic manner.

**WASP expression in macrophages regulates colitis.** To determine the macrophage- and DC-intrinsic role of WASP in mucosal homeostasis, we examined the impact of selective

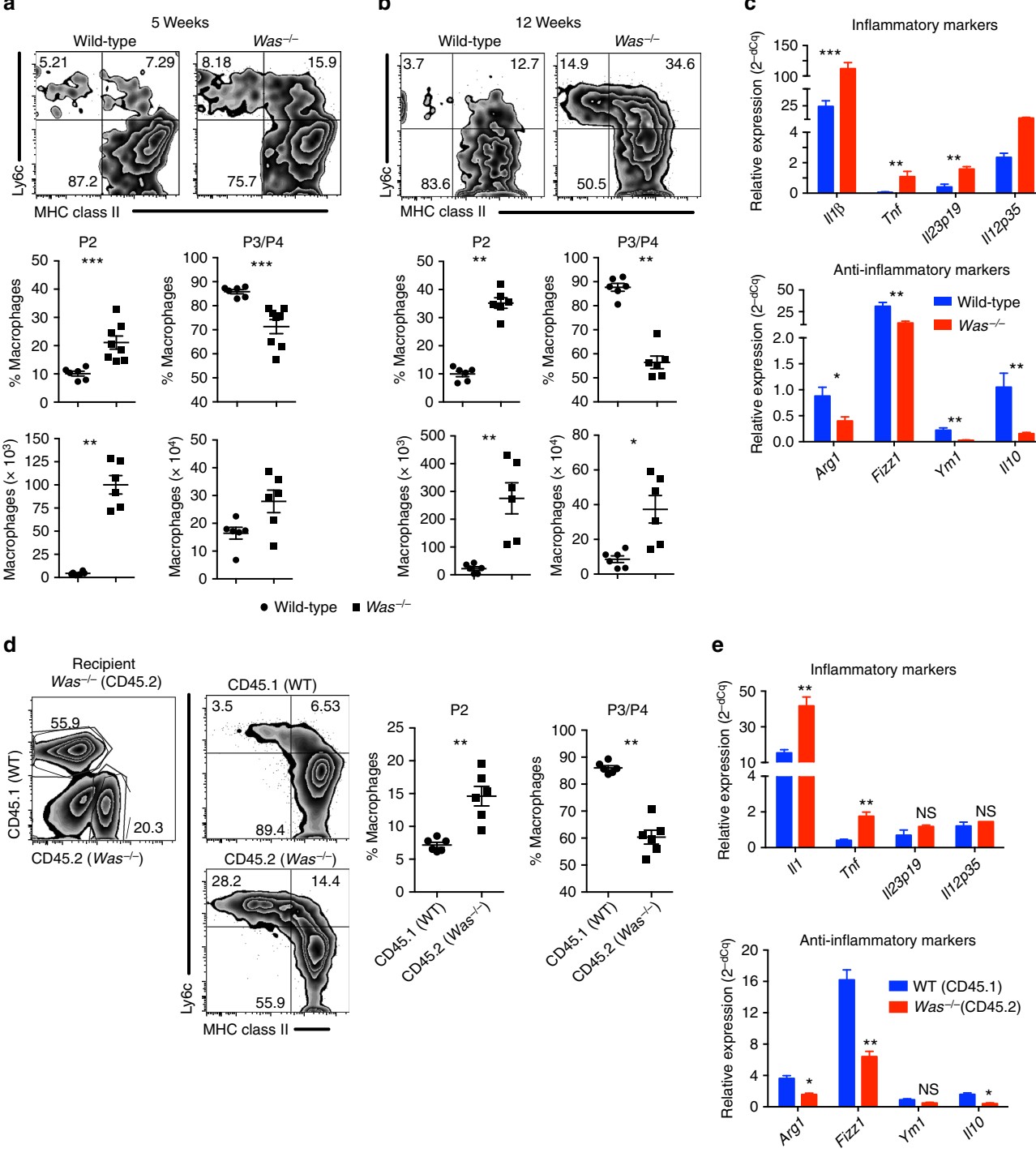

**Fig. 1** Defective anti-inflammatory macrophage differentiation and function in the colon of $Was^{-/-}$ mice. Flow cytometric analysis of LP macrophage in mice at **a** 5 (WT $n = 6$; $Was^{-/-}$ $n = 8$) and **b** 12 (WT $n = 6$; $Was^{-/-}$ $n = 6$) weeks of age followed by quantification of pro- (P2) and anti-inflammatory (P3 + P4) subsets. Macrophages were gated as live $CD45^+CD11b^+CD103^-CD64^+$ cells. Data are cumulative of three independent experiments. *$p < 0.05$, **$p < 0.01$, ***$p < 0.001$ (Student's t-test). **c** Expression of pro- and anti-inflammatory genes in sorted P3 + P4 macrophages (WT $n = 12$; $Was^{-/-}$ $n = 12$). P3 + P4 cells from three mice were pooled together. Data are cumulative of two independent experiments. *$p < 0.05$, **$p < 0.01$, ***$p < 0.001$ (Student's t-test). **d** $CD45.1^+$ (WT) and $CD45.2^+$ ($Was^{-/-}$) bone marrow cells were transferred at the ratio of 1:1 into lethally irradiated $CD45.2^+$ $Was^{-/-}$ recipient. LP macrophage was analysed after 10 weeks. FACS plot shows the gating strategy. Graph shows the quantification of P2 and P3/P4 cells in the WT ($n = 6$) and $Was^{-/-}$ ($n = 6$) compartment of recipient mice. Data are representative of two independent experiments. *$p < 0.05$, **$p < 0.01$, ***$p < 0.001$ (Student's t-test). **e** Expression of pro- and anti-inflammatory genes in sorted P3 + P4 macrophages in mice ($n = 8$). P3 + P4 cells from two mice were pooled together. *$p < 0.05$, **$p < 0.01$, ***$p < 0.001$, NS, not significant (Student's t-test). All graphs shows mean ± SEM

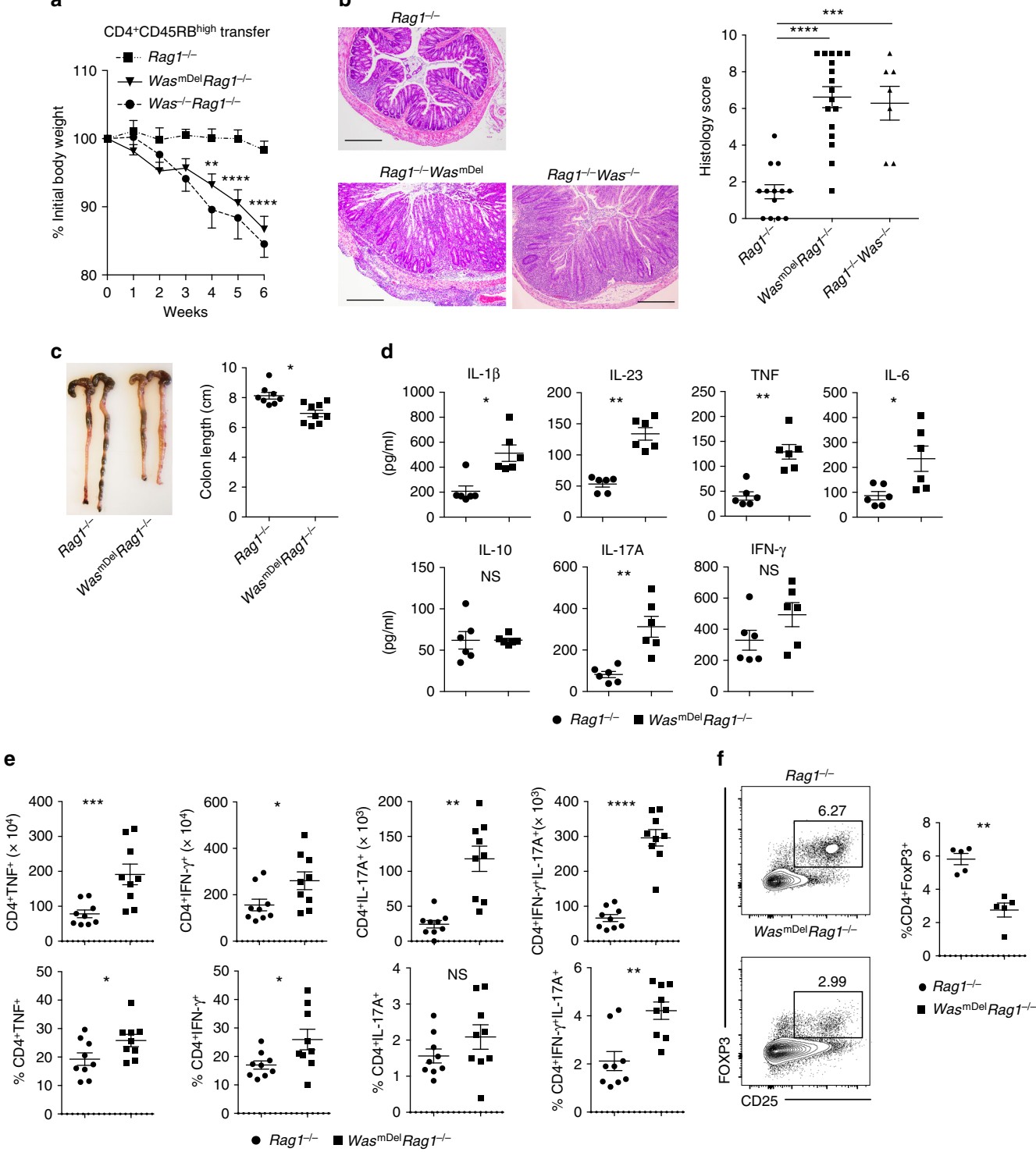

**Fig. 2** Macrophage-specific expression of WASP is critical for the regulation of T-cell-transfer-induced colitis. Naive CD4$^+$CD25$^-$CD45RB$^{hi}$ T cells (3–5 × 10$^5$) from wild-type (WT) mice were transferred i.p. into $Rag1^{-/-}$, $Was^{mDel}Rag1^{-/-}$ and $Was^{-/-}Rag1^{-/-}$ mice. **a** Mean ± SEM of percent initial body weight after transfer (% initial body weight). Data are cumulative of four independent experiments ($Rag^{-/-}$ $n = 13$, includes Exp 1, 2, 3, 4; $Was^{mDel}Rag1^{-/-}$ $n = 17$, includes Exp 1, 2, 3, 4, $Was^{-/-}Rag1^{-/-}$ $n = 7$, includes Exp 3, 4). Difference was not significant between $Was^{mDel}Rag1^{-/-}$ versus $Was^{-/-}Rag1^{-/-}$ cohorts. **$p < 0.01$, ****$p < 0.0001$ (two-way ANOVA). **b** Representative photomicrographs of H&E-stained colonic section and histological score after naive T-cell transfer. Scale bars: 200 μm. **c** Colon length at 6 weeks post transfer ($Rag^{-/-}$ $n = 8$; $Was^{mDel}Rag1^{-/-}$ $n = 9$). **d** Cytokines expression in colonic homogenates at 6 weeks post transfer. Data are cumulative of two independent experiments ($Rag^{-/-}$ $n = 6$; $Was^{mDel}Rag1^{-/-}$ $n = 6$). **e** Absolute number and frequency of TNF$^+$, IFN-γ$^+$, IL-17A$^+$ and IFN-γ$^+$IL-17A$^+$ helper T cells in the LP was determined by flow cytometry ($Rag^{-/-}$ $n = 9$; $Was^{mDel}Rag1^{-/-}$ $n = 9$). Data are cumulative of three independent experiments. **f** Percentage of Treg cells (CD45$^+$TCRβ$^+$CD4$^+$CD25$^+$FoxP3$^+$) in the LP was determined by flow cytometry ($Rag^{-/-}$ $n = 5$; $Was^{mDel}Rag1^{-/-}$ $n = 5$). Data are cumulative of two independent experiments. Data shown in **b–f** are mean ± SEM and $P$-value was obtained by Student's $t$-test. *$p < 0.05$, **$p < 0.01$, ***$p < 0.001$, ****$p < 0.0001$

deletion of WASP in macrophages and DCs. We first generated mice with macrophage- $Was^{fl/fl}LysM^{Cre}$ ($Was^{mDel}$) and DC- $Was^{fl/fl}Itgax^{Cre}$ ($Was^{dcDel}$) selective deletion of WASP on either the $Rag1^{-/-}$ or $Rag2^{-/-}$ background. Spontaneous inflammation was not observed in either $Was^{mDel}Rag1^{-/-}$ or $Was^{dcDel}Rag2^{-/-}$ mice at homeostasis. To evaluate whether WASP expression in

DCs contributes to disease pathogenesis, we transferred WT naive CD45RB[hi]CD4[+] T cells into either $Was^{dcDel}Rag2^{-/-}$ or $Rag2^{-/-}$ mice. After naive CD4[+] T cells transfer, both $Was^{dcDel}Rag2^{-/-}$ and $Rag2^{-/-}$ mice developed severe colitis (Supplementary Fig. 2b-2c). Only 10 weeks after transfer were significant differences in weight loss observed between $Was^{dcDel}Rag2^{-/-}$ and

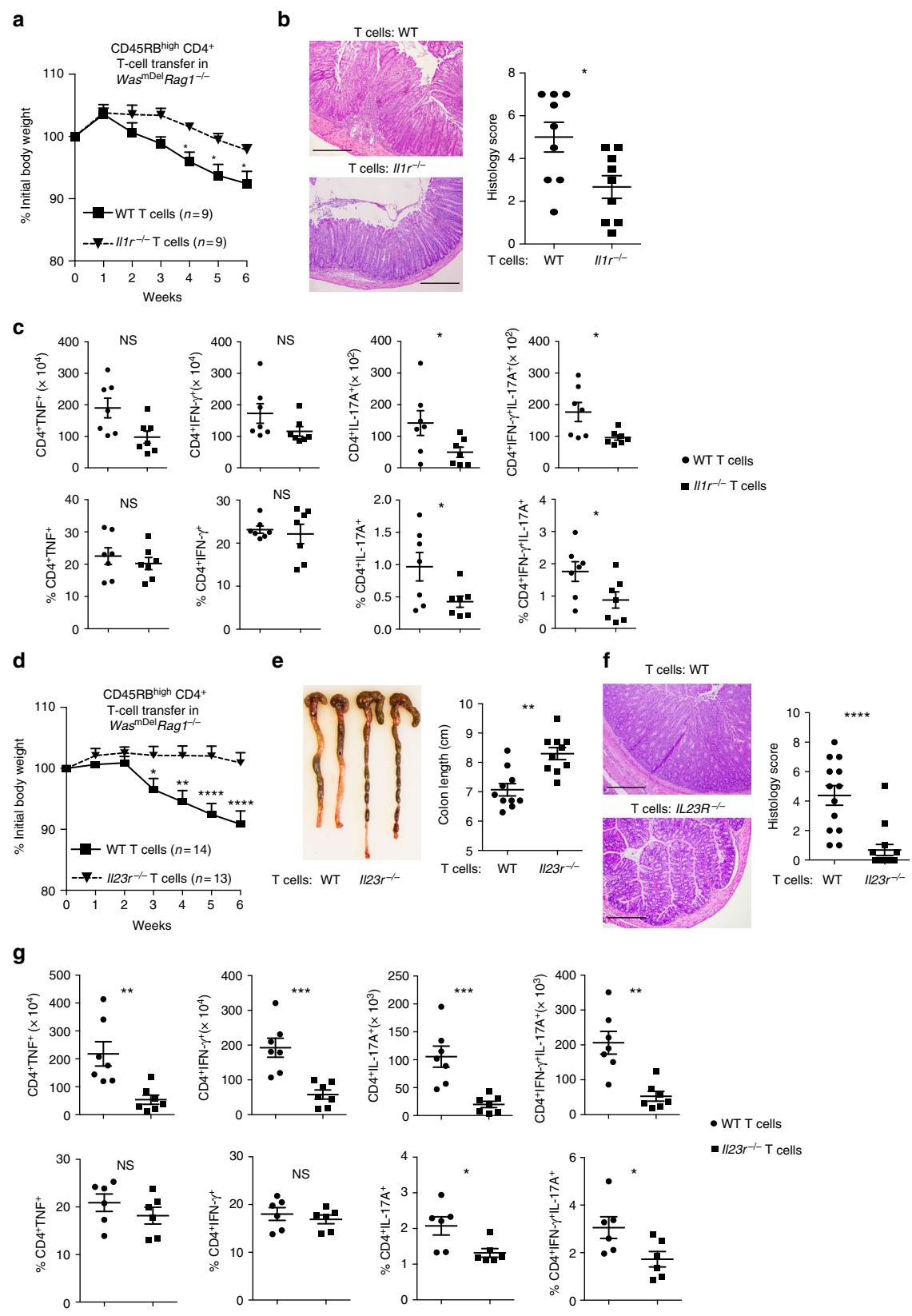

$Rag2^{-/-}$ recipient mice (*$p < 0.05$, Student's $t$-test) (Supplementary Fig. 2a). However, even at this late time point, colitis development was comparable in $Was^{dcDel}Rag2^{-/-}$ and $Rag2^{-/-}$ recipient mice (Supplementary Fig. 2c-2e).

In contrast to $Was^{dcDel}Rag2^{-/-}$ mice, $Was^{mDel}Rag1^{-/-}$ mice after transfer of WT naive $CD4^+$ T cells demonstrated increased weight loss as compared with $Rag1^{-/-}$ recipient controls. This weight loss was apparent as early as 4 weeks after transfer and was associated with increased inflammation (crypt abscesses, cellular infiltration and epithelial hyperplasia) and shortened colons (Fig. 2a-c). However, the disease is comparable between $Was^{mDel}Rag1^{-/-}$ and $Was^{-/-}Rag1^{-/-}$ mice (Fig. 2a, b). Expression of pro-inflammatory cytokines IL-1β, IL-23, tumor necrosis factor (TNF), IL-6 and IL-17 were markedly increased in $Was^{mDel}Rag1^{-/-}$ compared with $Rag1^{-/-}$ recipient mice (Fig. 2d). There was a trend toward higher expression of interferon (IFN)-γ in $Was^{mDel}Rag1^{-/-}$ mice colons; expression of IL-10 was comparable between groups. IL-1β, IL-23 and IL-6 are known drivers of Th17-type immune responses in the colonic mucosa[22–25]. Therefore, we compared IL-17-, INF-γ- and TNF-producing Th cells in the colon of $Rag1^{-/-}$ and $Was^{mDel}Rag1^{-/-}$ mice after naive T-cell transfer. We observed a marked increase in the absolute number of TNF, IL-17- and IFN-γ-producing Th cells, as well as an increase in IL-17-IFN-γ-double producing Th cells (Fig. 2e), which have been reported to be the most pathogenic subset[26,27]. The percentage of IL-17, IFN-γ and IL-17–IFN-γ double-positive cells were also increased in mesenteric lymph node (MLN) of $Was^{mDel}Rag1^{-/-}$ mice (Supplementary Fig. 3). Moreover, the in vivo generation of inducible Treg cells (iTregs) was also impaired in $Was^{mDel}Rag1^{-/-}$ mice, as was the percentage of $CD4^+CD25^+FoxP3^+$ cells in the colon and MLN 6 weeks after naive $CD4^+$ T-cell transfer compared with $Rag1^{-/-}$ recipient mice (Fig. 2f and Supplementary Fig. 3). Collectively, our data suggest that the expression of WASP in macrophages but not in DC is critical for the regulation of Th17/Th1-driven colitis after $CD4^+$ T-cell transfer.

**IL-1β and IL-23 drive pathogenesis in $Was^{mDel}Rag1^{-/-}$ mice.** As noted above, colonic macrophages produce higher amount of IL-1β and IL-23 in the absence of WASP (Fig. 1c). We also observed elevated expression of IL-1β and IL-23 in $Was^{mDel}Rag1^{-/-}$ recipient mice after transfer of naive $CD4^+$ WT T cells (Fig. 2d). We hypothesized that disease development in $Was^{mDel}Rag1^{-/-}$ mice after T-cell transfer is driven by macrophage-derived IL-1β and IL-23. To test the role of IL-1β in driving disease development, we transferred either WT or $Il1r^{-/-}$ naive T cells into $Was^{mDel}Rag1^{-/-}$ mice. Mice receiving $Il1r^{-/-}$ T cells had reduced weight loss, developed less colonic inflammation and showed significant reduction in IL-17 (*$p < 0.05$, Student's $t$-test) and IL-17-IFN-γ (*$p < 0.05$, Student's $t$-test) double-positive Th cells compared with those that received WT T cells (Fig. 3a-c). The frequency of IFN-γ$^+$ T cells and Tregs in recipients of either

WT or $Il1r^{-/-}$ naive CD4 T cells were similar (Fig. 3c and Supplementary Fig. 4a). Collectively, data suggests that IL-1β is a partial but not the sole driver of colitis in these mice.

Similarly, to assess the role of IL-23 in driving disease development, we transferred either $Il23r^{-/-}$ or WT naive T cells into $Was^{mDel}Rag1^{-/-}$ mice. Mice that received $Il23r^{-/-}$ naive T cells were completely protected form colitis compared with WT naive T-cell recipients with maintenance of body weight, colonic length and reduced colonic inflammation (Fig. 3d-f). There was a reduction in the absolute number of IL-17, IL-17–IFN-γ double-positive and IFN-γ-producing cells in the colon of $Was^{mDel}Rag1^{-/-}$ mice receiving $Il23r^{-/-}$ T cells compared with WT T cells (Fig. 3g). However, in vivo Treg generation was comparable between mice that received either $Il23r^{-/-}$ T cells or WT T cells (Supplementary Fig. 4b). These data support the hypothesis that macrophage-derived IL-23 and IL-1β drive disease development in $Was^{mDel}Rag1^{-/-}$ mice after T-cell transfer.

**WASP regulates M1 and M2 macrophages.** We sought to explore whether BMDMs differentiation and function is also dependent on WASP. BMDMs can be differentiated in vitro into either pro-inflammatory M1 or anti-inflammatory M2 macrophages using different combinations of polarizing agents. LPS (lipopolysaccharide) and IFN-γ treatment of BMDM generates M1-type macrophage, whereas the combination of IL-4, IL-13, TGF-β and IL-10 promotes the generation of M2-type macrophages[28,29]. Others and we reported the generation of highly immunosuppressive M2r type macrophages using a combination regimen of IL-4, TGF-β and IL-10[29,30]. These M2r macrophages express programmed death ligand 1 (PD-L1) and PD-L2, secrete IL-10 and TGF-β, suppress T-cell proliferation and are capable of preventing diabetes in NOD mice[29]. Compared with WT M2r, $Was^{-/-}$ M2r macrophages expressed lesser amount of M2-specific genes including $Arg1$, $Ym1$, $Fizz1$ and $Il10$ (Fig. 4a). The observed difference in M2 polarization in $Was^{-/-}$ mice is not due to any difference in the bone marrow progenitor population or in cultured BMDM prior to M2 differentiation (i.e., M0 cells), as they were comparable between WT and $Was^{-/-}$ mice both quantitatively and qualitatively. In the absence of TLR (toll-like receptor) stimulation, WT M2r macrophages express negligible amount of inflammatory mediators; upon LPS stimulation, WT M2r macrophages produce inflammatory cytokines but at significantly reduced levels when compared with M1 macrophages[30]. After restimulation with LPS, $Was^{-/-}$ M2r macrophages expressed higher amount of inflammatory cytokines including $Il1β$, $Il6$, $Tnf$ and $Il23$ compared with WT M2r macrophages (Fig. 4b). Moreover, functionally $Was^{-/-}$ M2r macrophages induced higher T-cell proliferation and induced less iTreg cell generation compared with WT M2r macrophages (Fig. 4c, d). Increased proliferation observed in presence of $Was^{-/-}$ M2r macrophages could be due to reduced expression of Arg1 as

**Fig. 3** Role of IL-1 and IL-23 signalling in T cells in the induction of colitis in $Was^{mDel}Rag1^{-/-}$ mice. Naive $CD4^+CD25^-CD45RB^{hi}$ T cells ($3–5 \times 10^5$) from WT or $Il1r^{-/-}$ mice were transferred i.p. into $Was^{mDel}Rag1^{-/-}$ mice. **a** Mean ± SEM % initial body weight (mean ± SEM) after transfer. (WT $n = 9$; $Il1r^{-/-}$ $n = 9$). *$p < 0.05$ (two-way ANOVA). Data are cumulative of two independent experiments. **b** Representative photomicrographs of H&E-stained colonic section and histological score at 6 weeks after transfer. Scale bars: 200 μm. **c** Absolute number and frequency of TNF$^+$, IFN-γ$^+$, IL-17A$^+$ and IFN-γ$^+$IL-17A$^+$ helper T cells in the LP was determined by flow cytometry (WT $n = 7$; $Il1r^{-/-}$ $n = 7$). Data are cumulative of two independent experiments. Naive CD4$^+$CD25$^-$CD45RB$^{hi}$ T cells ($3–5 \times 10^5$) from WT or $Il23r^{-/-}$ mice were transferred i.p. into $Was^{mDel}Rag1^{-/-}$ mice. **d** Mean ± SEM % initial body weight (mean ± SEM) after transfer. (WT $n = 14$; $Il23r^{-/-}$ $n = 13$). *$p < 0.05$, **$p < 0.01$, ****$p < 0.0001$ (two-way ANOVA). Data are cumulative of three independent experiments. **e** Colon length at 6 weeks post transfer. (WT $n = 10$; $Il23r^{-/-}$ $n = 10$). **f** Representative photomicrographs of H&E-stained colonic section and histological score at six weeks after transfer. Scale bars: 200 μm. **g** Absolute number and frequency of cytokine producing helper T cells in the LP was determined by flow cytometry (WT $n = 7$; $Il23r^{-/-}$ $n = 7$). Data are cumulative of two independent experiments. Data shown in **b,c** and **e–g** are mean ± SEM and $P$-value was obtained by Student's $t$-test. *$p < 0.05$, **$p < 0.01$, ***$p < 0.001$, ****$p < 0.0001$

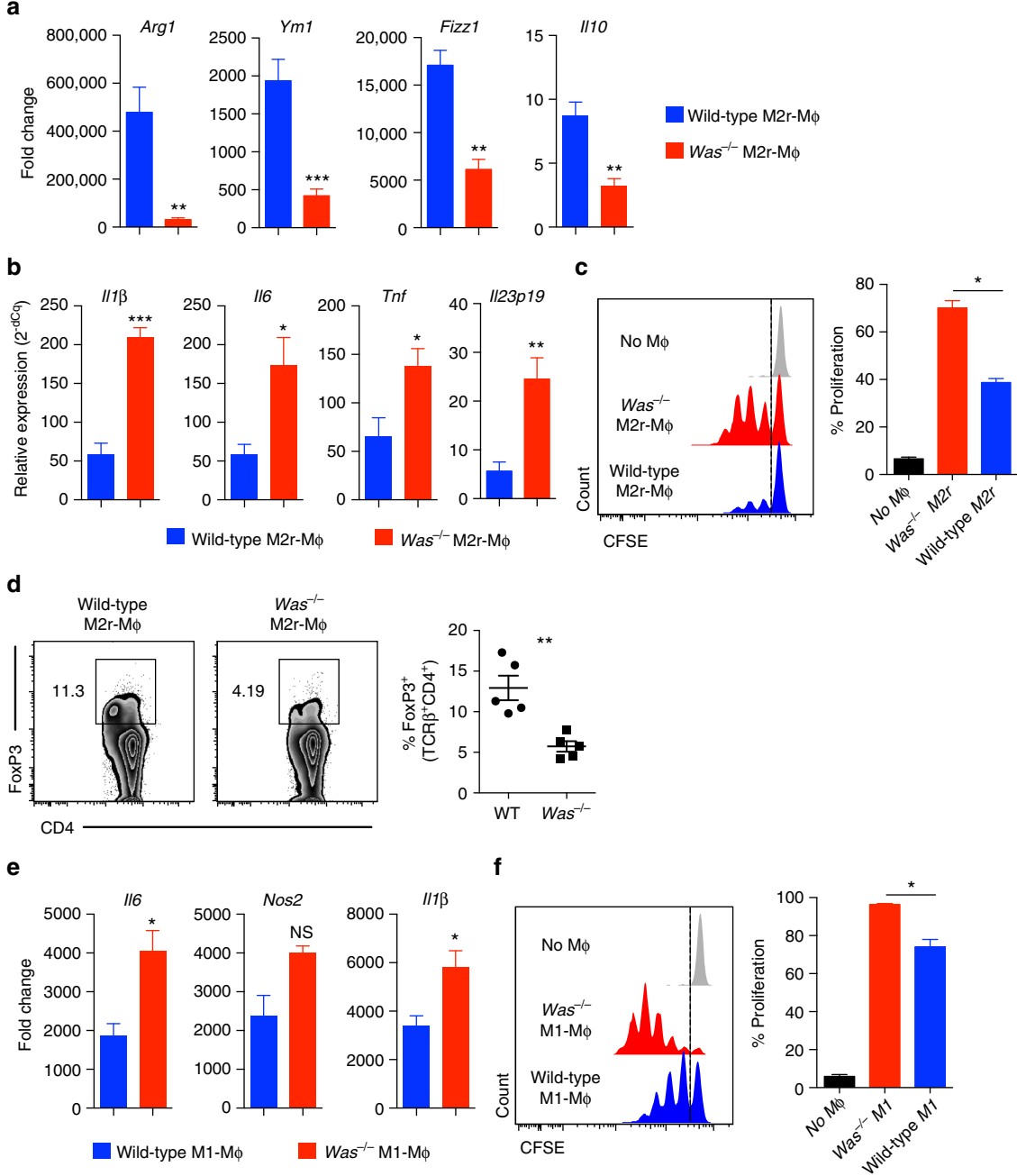

**Fig. 4** Defective in vitro differentiation and function of M2r and M1 macrophage in absence of WASP. BMDM from WT and $Was^{-/-}$ mice were cultured in presence of IL-4, IL-10 and TGF-β to differentiate them into M2r macrophages. **a** qPCR analysis of M2 specific gens expression (WT $n = 5$; $Was^{-/-}$ $n = 5$). Data are representative of three independent experiments. **b** qPCR analysis of proinflammatory gene expression in WT ($n = 5$) and $Was^{-/-}$ ($n = 5$) M2r-macrophage after restimulation with LPS for 4 h. Data are representative of three independent experiments. **c** CFSE (carboxyfluorescein succinimidyl ester)-labelled naive CD4+CD25− T cells were cultured in presence of plate bound anti-CD3 (2μg/ml) and either WT or $Was^{-/-}$ M2r macrophages for four days. T-cell proliferation was determined by flow cytometer. Data are representative of three independent experiments. **d** Naive CD4+CD25− T cells were cultured in the presence of plate-bound anti-CD3 (2 μg/ml), TGF-β (2 ng/ml) and either WT or $Was^{-/-}$ M2r macrophages for 5 days. FoxP3 expression were analysed by flow cytometry and quantified. Data are representative of three independent experiments. BMDM from WT and $Was^{-/-}$ mice were cultured in presence of LPS and INFγ to differentiate them into M1 macrophages. **e** qPCR analysis of M1 specific gens expression (WT $n = 5$; $Was^{-/-}$ $n = 5$). Data are representative of three independent experiments. **f** CFSE-labelled naive CD4+CD25- T cells was cultured in the presence of plate-bound anti-CD3 (2μg/ml) and WT or $Was^{-/-}$ M1 macrophage for 3 days. T-cell proliferation was determined by flow cytometer. Data are representative of three independent experiments. Data shown in **a–f** are mean ± SEM and *P*-value was obtained by Student's *t*-test. *$p < 0.05$, **$p < 0.01$, ***$p < 0.001$

described by Van de Velde et al.[31]. In M1 polarizing conditions, $Was^{-/-}$ macrophages express higher amounts of M1-specific genes including *Il6*, *Nos2* and *Il1β* in comparison with WT macrophages (Fig. 4e). $Was^{-/-}$ M1 macrophages, upon co-

culture with naive CD4+ T cells, induced higher T-cell proliferation compared with WT M1 macrophages (Fig. 4f). Collectively, these data indicate that WASP is critical for the differentiation and function of tolerogenic BMDM.

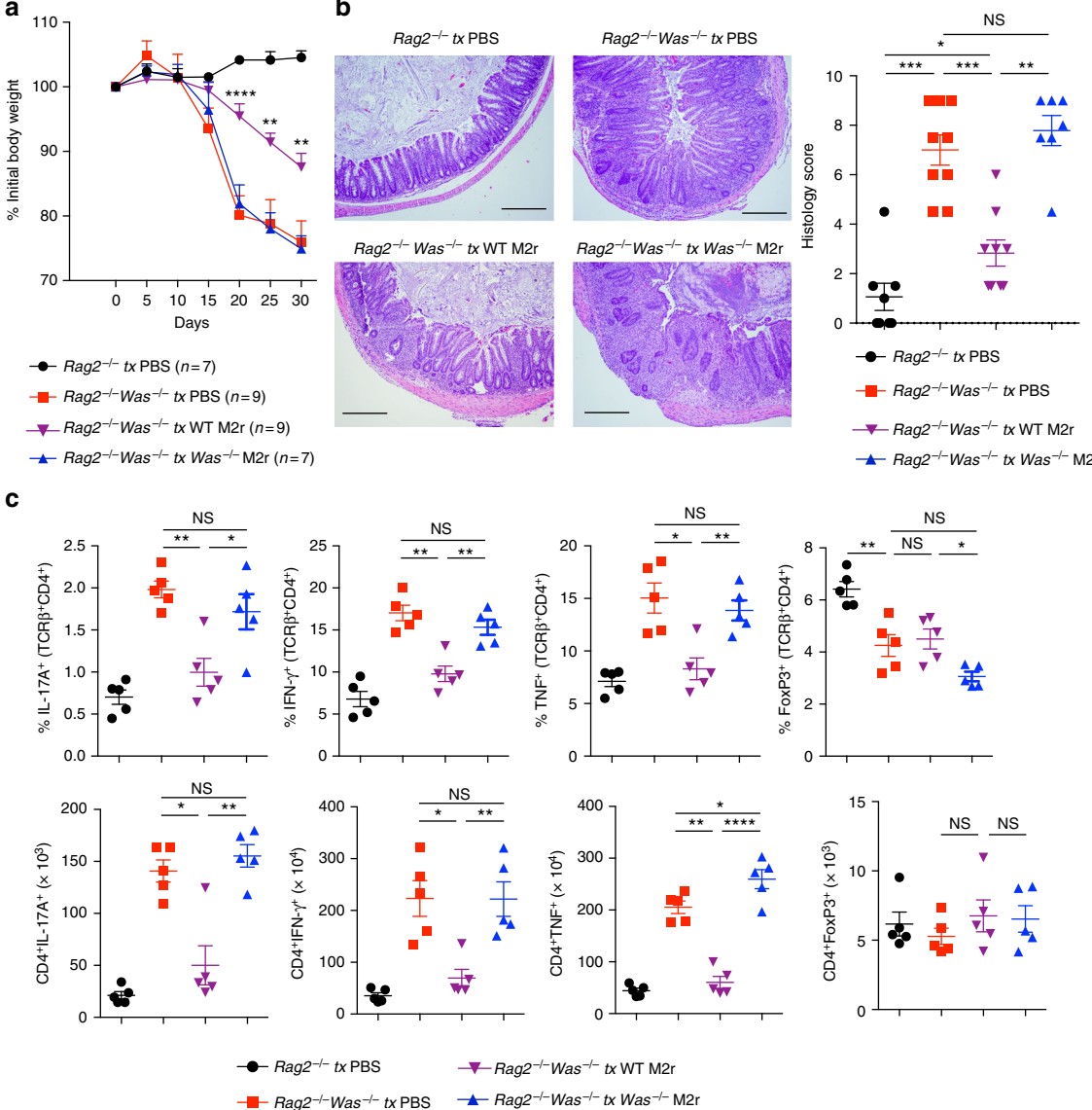

**Fig. 5** Wild-type M2r macrophage protect $Was^{-/-}Rag^{-/-}$ mice from T-cell-transfer-induced colitis. $Was^{-/-}Rag1^{-/-}$ mice were transfer with PBS ($n = 9$), WT M2r ($2 \times 10^6$) ($n = 9$) or $Was^{-/-}$ M2r ($2 \times 10^6$) ($n = 7$) macrophages one-day before WT CD4$^+$ T-cell transfer. As a control $Rag1^{-/-}$ ($n = 7$) were treated with PBS and transferred with WT CD4$^+$ T cells. **a** Mean ± SEM % initial body weight after T-cell transfer. Data are cumulative of two independent experiments. **$p < 0.01$, ***$p < 0.001$ (two-way ANOVA) ($Was^{-/-}Rag1^{-/-}$ tx PBS Vs $Was^{-/-}Rag1^{-/-}$ tx WT M2r). **b** Representative photomicrographs of H&E stained colonic section and histological score. Scale bars: 200 μm. **c** Absolute number and frequency of cytokine producing helper T cells in the LP was determined by flow cytometry. ($Rag1^{-/-}$ tx PBS $n = 5$; $Was^{-/-}Rag1^{-/-}$ tx PBS $n = 5$; $Was^{-/-}Rag1^{-/-}$ tx WT M2r $n = 5$; $Was^{-/-}Rag1^{-/-}$ tx $Was^{-/-}$ M2r $n = 5$). Data are cumulative of two independent experiments. Data shown in **b**,**c** are mean ± SEM and P-value was obtained by Student's t-test. *$p < 0.05$, **$p < 0.01$, ***$p < 0.001$, ****$p < 0.0001$; NS, not significant; tx: treated

**M2r macrophages protect $Was^{-/-}Rag2^{-/-}$ mice from colitis.** Taken together, our in vivo and in vitro data suggest that macrophages lose their tolerogenic function in the absence of WASP, thereby inducing robust effector T-cell expansion and colitis. We wanted to explore whether adoptive transfer of tolerogenic M2r macrophages could prevent T-cell transfer induced colitis in $Was^{-/-}Rag2^{-/-}$ mice. We previously reported that $Was^{-/-}Rag2^{-/-}$ mice when transferred with unfractioned total CD4$^+$ T cells develop severe colonic inflammation within 3 weeks[12]. To evaluate the function of M2r macrophages, we transferred either WT or $Was^{-/-}$ M2r BMDM 1 day before CD4$^+$ T-cell transfer and confirmed their presence in the LP after 7 days (Supplementary Fig. 5a). After CD4$^+$ T-cell transfer, $Was^{-/-}Rag2^{-/-}$ mice treated with WT M2r BMDM lost less body weight compared with mock-treated $Was^{-/-}Rag2^{-/-}$ mice (Fig. 5a). In contrast,

$Was^{-/-}$ M2r BMDM failed to protect $Was^{-/-}Rag2^{-/-}$ mice from weight loss. Histopathology showed significant reduction in colonic inflammation in $Was^{-/-}Rag2^{-/-}$ mice (**$p < 0.01$, ***$p < 0.001$, Student's t-test) treated with WT M2r BMDM but not with $Was^{-/-}$ M2r BMDM (Fig. 5b). Evaluation of colonic Th cells showed a reduction in percentage of IL-17A, IFN-γ and TNF-producing cells in WT M2r BMDM treated compared with mock-treated $Was^{-/-}Rag2^{-/-}$ mice (Fig. 5c). We did not observe an increase in the percentage and number of Tregs in $Was^{-/-}Rag2^{-/-}$ mice transferred with WT M2r macrophages (Fig. 5c). Colonic tissue expression of inflammatory genes was also reduced in $Was^{-/-}Rag2^{-/-}$ mice treated with WT M2r macrophages (Supplementary Fig. 5). These results indicate that colitis development in $Was^{-/-}Rag2^{-/-}$ mice can be improved by restoring tolerogenic macrophage population.

**IL-10 signalling is impaired in *Was*⁻/⁻ macrophages.** Differentiation of macrophages into anti-inflammatory M2-type cells is predominantly driven by transcription factors signal transducer and activator of transcription 6 (STAT6) and STAT3 downstream of IL-4, IL-13 and IL-10 signalling[29,30,32]. As described above, M2r polarization of *Was*⁻/⁻ macrophages was impaired in the presence of IL-4, IL-10 and TGF-β. The expression of IL-10R, IL-4R and TGF-βR was comparable between WT and *Was*⁻/⁻

macrophages at homeostasis (Supplementary Fig. 6a and d). Given the role of these cytokine in macrophages differentiation, we hypothesized that WASP may have a role in the regulation of IL-4 or IL-10 signalling, or both. Upon evaluation of IL-4 signalling in BMDM we did not find any difference in STAT6 phosphorylation between WT and *Was*⁻/⁻ BMDM, indicating functional IL-4 signalling (Fig. 6a and Supplementary Fig. 6b). To assess the role of WASP in IL-10-mediated STAT3

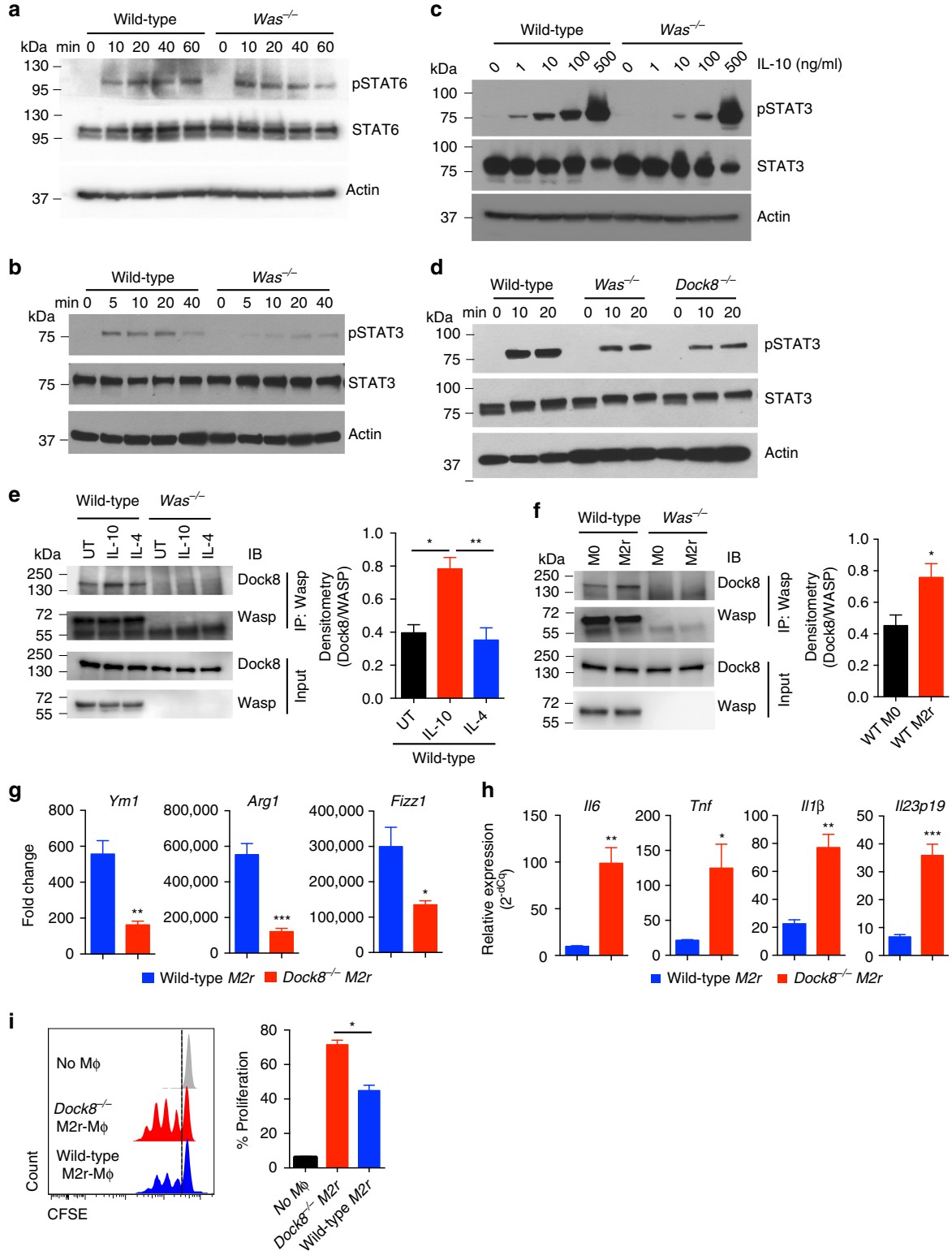

phosphorylation, we treated BMDM with IL-10 and analysed STAT3 phosphorylation at different time points. IL-10-mediated STAT3 phosphorylation was delayed and reduced in $Was^{-/-}$ BMDM (Fig. 6b). However, the defect in STAT3 phosphorylation was restricted to lower concentrations of IL-10, as we could rescue STAT3 phosphorylation at increased IL-10 concentrations (Fig. 6c). In addition, we found that the differential expression of M2-specific markers in $Was^{-/-}$ M2r macrophages is predominantly due to defect(s) in IL-10 signalling (Supplementary Fig. 6c). Expression of IL-10 target genes, including IL-4R and SOCS3, was also significantly reduced in $Was^{-/-}$ macrophages (**$p < 0.01$, ***$p < 0.001$, Student's $t$-test) (Supplementary Fig. 6d and e). Taken together, our data indicate that WASP is involved in IL-10-mediated STAT3 phosphorylation and suggest that defective STAT3 activation after IL-10R binding may be responsible for aberrant M2 macrophages polarization.

**WASP-DOCK8 forms a complex downstream of IL-10 signalling.** Recent studies have demonstrated that DOCK8 is involved in the regulation of STAT3 phosphorylation downstream of TLR9 signalling in B cells[33], IL-23R signalling in innate lymphoid cells[34] and IL-6R signalling in T cells[35]. Moreover, DOCK8 was recently shown to directly interact with STAT3[35]. DOCK8 also constitutively interacts with WASP through WIP (WASP-interacting protein) in T cells[36]. We hypothesized that WASP regulates IL-10-mediated STAT3 phosphorylation through an interaction with DOCK8. We found that IL-10-mediated STAT3 phosphorylation is also reduced in $Dock8^{-/-}$ compared with WT BMDM (Fig. 6d). To examine whether WASP and DOCK8 form a complex during IL-10 signalling, we immunoprecipitated WASP from untreated, IL-10, IL-4-treated BMDM. We found that WASP interacts with DOCK8 constitutively and the interaction is enriched in the presence of IL-10 and also after M2r conditioning (Fig. 6e, f). To further assess the role of DOCK8 in M2r differentiation, we differentiated $Dock8^{-/-}$ BMDM into M2r macrophages. Expression of M2-specific genes including $Arg1$, $Ym1$ and $Fizz1$ was lesser in $Dock8^{-/-}$ compared with WT BMDM (Fig. 6g). Similar to $Was^{-/-}$ M2r macrophages, restimulation of $Dock8^{-/-}$ M2r macrophages with LPS, was associated with higher expression of inflammatory genes including $Il6$, $Tnf$, $Il1\beta$ and $Il23p19$ (Fig. 6h). In a T-cell co-culture assay, $Dock8^{-/-}$ M2r macrophages induced higher T-cell proliferation compared with WT (Fig. 6i). Together, these results demonstrate that similar to WASP, DOCK8 also regulates anti-inflammatory macrophages differentiation, and that IL-10 signalling induces a WASP/DOCK8 complex.

**Defective macrophage function in WAS patients.** We next sought to investigate whether patients with WAS also exhibit defects in macrophage polarization and function similar to $Was^{-/-}$ mice. CD14$^+$ monocytes from periheral blood mononuclear cells (PBMCs) of seven patient and matched healthy control were differentiated into macrophages and polarized to M1 or M2r macrophages. Similar to murine $Was^{-/-}$ M1 macrophages, M1 macrophages from WAS patients expressed higher quantities of M1-specific transcripts including $CXCL10$ and $CCR7$ compared with M1 macrophages from healthy controls (Fig. 7a). Furthermore, in an in vitro macrophage T-cell co-culture assay, WASP-deficient human M1 macrophages induced enhanced T-cell proliferation when compared with control M1 macrophages (Fig. 7b).

In contrast, expression of human M2-specific genes including $CCL13$, $SLC38A6$, $MRC1$, $CXCR4$ and $IL10$ was markedly reduced in six out of seven patients compared with healthy controls (Fig. 7c). In addition, when M2 differentiated macrophages were re-stimulated with LPS, WAS patient macrophages showed higher expression of proinflammatory genes including $IL6$, $TNF$, $IL23$ and $IL1\beta$ analogous to findings in murine $Was^{-/-}$ M2r BMDM (Fig. 7d). Co-culture of control or WAS patient M2r macrophages with naïve T cells showed higher T-cell proliferation in the presence of WAS patient-derived M2r macrophages compared with control, indicating that human WASP-deficient M2r macrophages have higher pro-inflammatory potential (Fig. 7e). Moreover, M2r macrophages derived from WAS-deficient patients induced less iTreg cell generation compared with control M2 macrophages when co-cultured with naive CD4$^+$CD25$^-$ T cells (Fig. 7f). Furthermore, WAS patient-derived M2r macrophages upon co-culture with naive T cells induces more TNF production and less IL-10 production by T cells (Fig. 7g), which infers that the T cells in the presence of WASP-deficient M2r macrophages develops into more effector-type rather then regulatory-type T-helper cells. Collectively, our data suggest that the generation and function of tolerogenic macrophages requires intact WASP signalling both in mice and human.

**Discussion**

Tissue resident macrophages have an important role in the maintenance of immune tolerance. In the intestinal LP, resident anti-inflammatory macrophages regulate mucosal homeostasis at least in part through the generation of regulatory T cells and suppression of T-cell proliferation. Alterations in macrophage function in the intestine, due to genetic or environmental factors, can lead to abnormal activation of innate and adaptive immune responses, and result in intestinal inflammation. Recently, several groups including our own have reported that dysfunctional anti-inflammatory macrophages in the absence of IL-10 signalling result in intestinal inflammation[30,37,38]. An IBD focused network analysis predicted that IL-10 and other molecules including

**Fig. 6** STAT3 phosphorylation downstream of IL-10R signalling involves WASP-DOCK8 complex. WT and $Was^{-/-}$ BMDM were treated with IL-4 (20ng/ml) **a** or IL-10 (20ng/ml) **b** for the indicated time and pSTAT6 or pSTAT3 was analysed by western blotting, respectively. Data are representative of three independent experiments. **c** WT and $Was^{-/-}$ BMDM were treated with various concentration of IL-10 for 20 min and pSTAT3 was analysed by western blotting. Data are representative of three independent experiments. **d** WT, $Was^{-/-}$ and $Dock8^{-/-}$ BMDMs were treated with IL-10 for the indicated time and pSTAT3 was analysed by western blotting. Data are representative of three independent experiments. **e** BMDM were treated with IL-10 (20ng/ml) or IL-4 (20ng/ml) for 30 min. Cell extracts were immunoprecipitated (IP) with anti-WASP antibody and interaction with DOCK8 is detected with immunoblotting (IB). Data are representative of three independent experiments with similar results and the histogram shows the cumulative densitometry. **f** Cell extracts from M0 and M2r macrophages were immunoprecipitated with anti-WASP antibody and interaction with DOCK8 is detected with immunoblotting. Data are representative of three independent experiments with similar results and the histogram shows the cumulative densitometry. BMDM from WT and $Dock8^{-/-}$ mice were cultured in the presence of IL-4, IL-10 and TGF-β for 24 h to differentiate them into M2r macrophages. **g** qPCR analysis of M2-specific genes in WT ($n = 5$) and $Dock8^{-/-}$ ($n = 5$) M2r-BMDM. **h** Expression of proinflammatory genes by WT ($n = 5$) and $Dock8^{-/-}$ ($n = 5$) M2r-BMDM after restimulation with LPS for 4 h was determined by qPCR. Data are representative of three independent experiments **g**,**h**. **i** CFSE-labelled naive CD4$^+$CD25$^-$ T cells were cultured in the presence of plate bound anti-CD3 (2μg/ml) and either WT or $Dock8^{-/-}$ M2r macrophages for 4 days. T-cell proliferation was determined by flow cytometer. Data are representative of three independent experiments. Data shown in **e**–**i** are mean ± SEM and P-value was obtained by Student's t-test. *$p < 0.05$, **$p < 0.01$, ***$p < 0.001$

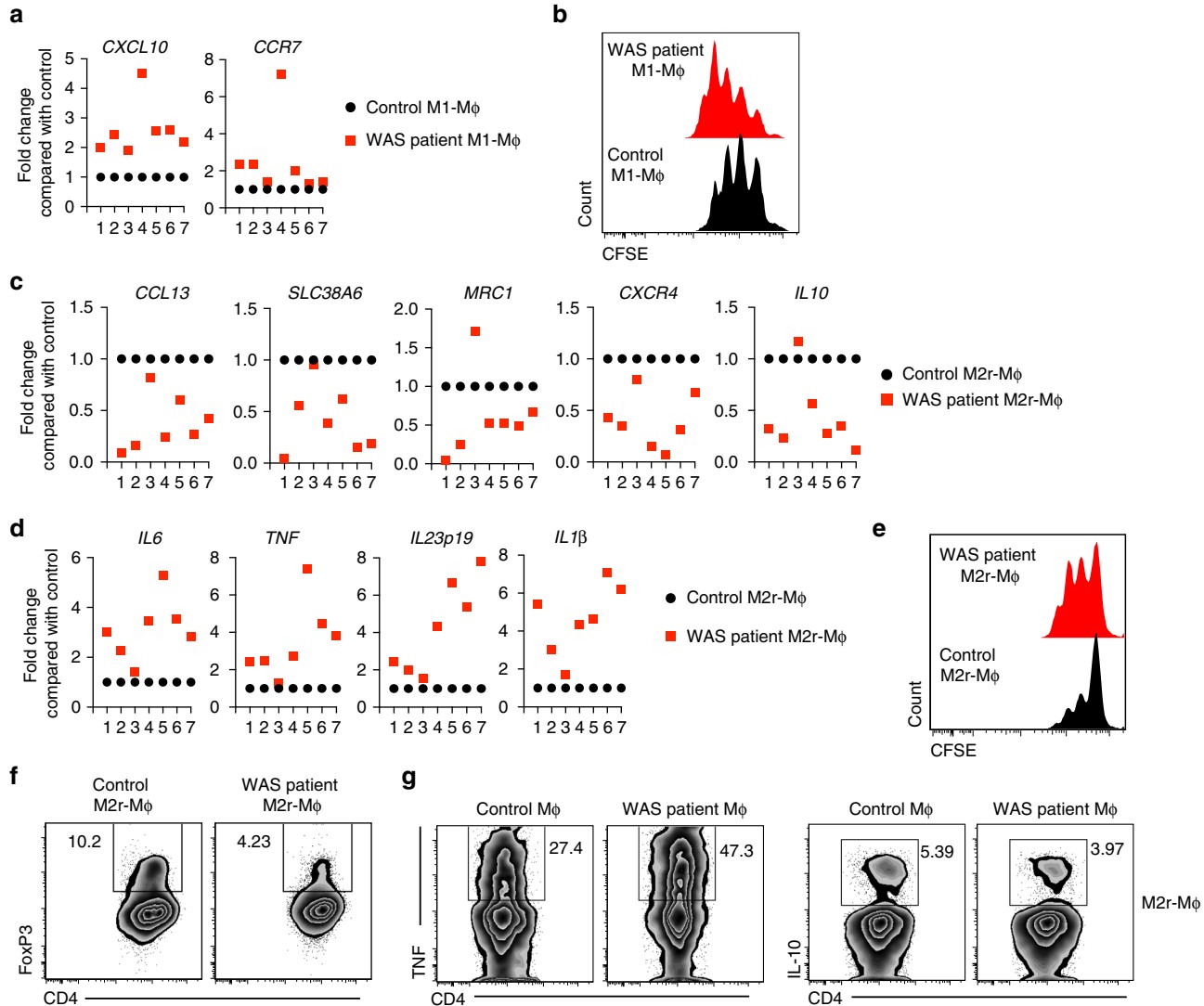

**Fig. 7** Macrophage differentiation and function is impaired in patients with WAS. Monocyte-derived macrophages form healthy donors (control) and patients with WAS were differentiated into M1 macrophages. **a** Expression of M1-specific markers was analysed by qPCR. **b** Naive CD4$^+$CD25$^-$ T cells were CFSE labelled and cultured in the presence of plate-bound anti-CD3 (2 μg/ml) and either control or WAS patient-derived M1 macrophages for 3 days. T-cell proliferation was determined by flow cytometer. Data are representative of two independent experiments with two patient samples. Monocyte derived macrophages form healthy donors (control) and patients with WAS were differentiated into M2r macrophages. **c** Expression of M2-specific markers was analysed by qPCR. **d** M2r differentiated monocyte-derived macrophages were restimulated with LPS for 4 h. Expression of pro-inflammatory genes were analysed by qPCR. **e** Naive CD4$^+$CD25$^-$ T cells were CFSE labelled and cultured in the presence of plate-bound anti-CD3 (2 μg/ml) and either control or WAS patient-derived M2r macrophages for 4 days. T-cell proliferation was determined by flow cytometer. Data are representative of two independent experiments with two patient samples. **f** Naive CD4$^+$CD25$^-$ T cells were cultured in presence of plate-bound anti-CD3 (2μg/ml), TGF-β (2 ng/ml) and either control or WAS patient-derived M2r macrophages for 5 days. Treg generation was examined by analysing FOXP3 expression using flow cytometry. Data are representative of two independent experiments with two patient samples. **g** Naive CD4$^+$CD25$^-$ T cells from independent donor were cultured in the presence of plate-bound anti-CD3 (2 μg/ml) and either control or WAS patient-derived M2r macrophages for 3 days. Cells were restimulated with PMA (10 ng/ml), ionomycin (500 ng/ ml) in the presence of GogiStop for last 4 h. Intracellular cytokine expression was analysed by flow cytometry. Data are representative of two independent experiments with two patient samples

WASP and NOD2, enriched in anti-inflammatory macrophages, act in concert to regulate intestinal immune homeostasis[1]. Although WASP has been described to have a key role in lymphocyte function[5,39], we have previously demonstrated that WASP can also regulate innate immune cell function and intestinal homeostasis[12]. Here we investigated the role of myeloid cell-specific WASP expression in intestinal homeostasis and postulated that WASP regulates macrophages tolerogenic function.

Although WASP modulates diverse macrophage functions[40,41], the role of WASP in the regulation of anti-inflammatory function of macrophages in the context of intestinal homeostasis has not been explored. We demonstrate that WASP facilitates the differentiation of circulating monocytes into anti-inflammatory tissue resident macrophages. We observed the accumulation of inflammatory P2 macrophages and reduction in anti-inflammatory P3/P4 macrophages within the LP in the absence of WASP. The expressions of inflammatory cytokines were also elevated in P3/P4 macrophages in $Was^{-/-}$ mice. A key point in these studies is that the absence of WASP not only causes a reduction in the frequency of regulatory macrophages but it also dampened the regulatory potential of P3/P4 macrophages. Using

both $Was^{mDel}Rag1^{-/-}$ and $Was^{dcDel}Rag2^{-/-}$ mice we found that WASP expression in macrophages but not in DCs was essential to restrict colitis development. We further identified that deletion of WASP in macrophages upregulates expression of multiple pro-inflammatory cytokines in association with expansion of pathogenic Th1/Th17 cells and disease exacerbation.

Numerous studies have identified a role of IL-1β and IL-23 in promoting Teff cell mediated diseases in both human and mice[23,24,26,42]. Previously, we reported elevated colonic expression of IL-1β and IL-23 in mice lacking WASP in all innate immune cells[12]. Here we demonstrate that macrophage specific deletion of WASP is sufficient to drive enhanced expression IL-1β and IL-23, which promotes generation of pathogenic IL-17[+]IFN-γ[+] CD4[+] T cells and colitis. In addition, our study, demonstrates that colitis development in the absence of WASP requires both IL-1 and IL-23. Similar to our finding, Krause et al.[43] reported that enhanced macrophage-derived IL-23, in the absence of myeloid-specific IL-10, causes increased mortality in infectious colitis model. Our work complements a recently reported human study that describes a WAS patient with autoinflammatory manifestations, pan colitis and perianal abscesses, who showed improvement after treatment with an IL-1 receptor antagonist[44].

Several prior studies have shown that WASP plays an important role in Treg function[45–47]. Here we observe that aberrant macrophage differentiation and function associated with selective WASP deficiency in macrophages affects Treg cell generation. Previously we demonstrated a decrease in iTreg cell generation in $Was^{-/-}Rag2^{-/-}$ mice transferred with CD4[+] T cells[12]. Our result using $Was^{mDel}Rag1^{-/-}$ mice re-enforced these findings suggesting that the defect in iTreg generation may be due, at least in part, to defects in macrophage function. This reduction in Treg cell generation could be due to elevated colonic expression of IL-23 as described by Ahern et al.[26]. Together we can conclude that the elevated expression of inflammatory mediators by WASP-deficient macrophages causes expansion of pathogenic Teff cells and reduction of Treg cells, which ultimately lead to breakdown of immune tolerance.

Similar to in vivo observations in the colon, the generation and function of M1 and M2r macrophages from BM was also aberrant in WASP deficiency and restoration of anti-inflammatory macrophages partially rescued colonic inflammation in $Was^{-/-}Rag2^{-/-}$ mice after T-cell transfer. Prior studies have reported amelioration of inflammation in several models of colitis with anti-inflammatory macrophages[15,20]. These observations support our findings that macrophages have (WASP-dependent) tolerogenic properties. Importantly, analogous to our findings in mice, we observed defects in the generation and function of anti-inflammatory macrophages in WAS patients. Monocyte-derived M2r macrophages from WAS patients showed aberrant tolerogenic properties. These data suggest that polymorphisms that impact WASP expression may play a role in tolerogenic macrophage function and intestinal homeostasis. Indeed, about 10% of patients with WAS develop IBD[48]. Together, our results implicate a critical role for WASP in macrophages in regulating mucosal tolerance in both mice and human.

Many studies have shown the involvement of STAT6- and STAT3-dependent signalling pathways in regulatory macrophage function. We recently reported that IL-10-mediated STAT3 phosphorylation is critical for regulatory macrophage function and intestinal homeostasis[30]. Moreover, selective deletion of STAT3 in macrophages causes spontaneous colitis in mice[15]. Here we identify a mechanistic role for WASP in specifically regulating IL-10-dependent STAT3 phosphorylation. DOCK8, a guanine nucleotide exchange factor, known to interact with WASP[49], has been recently reported to regulate STAT3 phosphorylation in multiple cell types (including B cells, innate

lymphoid cells and T cells) downstream of TLR9-, IL-23- and IL-6-dependent signals[33–35], and our data specifically links WASP to STAT3 phosphorylation downstream of IL-10 signalling in macrophages. The interaction between DOCK8 and WASP in T cells appears to be mediate through[36] WIP. Our studies indicate that WASP and DOCK8 also interact in macrophages and that this interaction is enhanced by IL-10-stimulation. Defective macrophage differentiation and IL-10-dependent STAT3 phosphorylation in DOCK8-deficient macrophages further strengthens our conclusion that WASP, together with DOCK8, regulates anti-inflammatory macrophage function and protects from colitis development. Although IL-10-dependent signalling induces the formation of a WASP-DOCK8 signalling complex, further experimentation is needed to delineate the role of a WASP-DOCK8 complex in the regulation of IL-10 signalling. Report of IBD patients with polymorphisms in DOCK8 gene[50] supports our notion of involvement of a WASP-DOCK8 signalling axis in the regulation of tolerogenic macrophage function and intestinal homeostasis.

In summary we have demonstrated that macrophage-selective expression of WASP is critical for the development of anti-inflammatory functions by LP and BM-derived macrophages. In WASP-deficient macrophages, expression of inflammatory cytokines is elevated leading to exacerbated Th1/Th17-helper cell response and abrogated iTreg generation. Many of these phenotype observed in $Was^{mDel}Rag1^{-/-}$ mice are similar to what we reported in mice with innate immune cell-specific deletion of IL-10R[30]. Similar to our finding in IL-10R-deficient patients, our current study shows generation and function of tolerogenic M2r macrophages is also defective in WAS-deficient patients. WAS-deficient patients can present with phenotype similar to IL-10R deficient patients including recurrent infections, colitis and perianal disease[44,51]. This similar phenotype may be explained, at least in part, by our finding that IL-10 signalling is abnormal in WASP deficient macrophages. Collectively, our data suggests that aberrant anti-inflammatory macrophage function and mucosal tolerance in WASP-deficient mice and humans may be mediated by inappropriate IL-10-mediated STAT3 signalling. All in all, our study posits that manipulation of macrophage function, IL-10 signalling and/or IL-1/IL-23-mediated effector T-cell function may have therapeutic benefit in WAS-deficient patients.

## Methods

**Mice.** C57BL/6 background $Was^{fl/fl}$ mice have been described previously[52] and were crossed with $LysM^{Cre}$ (The Jackson Laboratory, Stock No. 004781) or $Itgax^{Cre}$ (The Jackson Laboratory, Stock No. 008068) mice obtained from Jackson Laboratories to generate $Was^{fl/fl}LysM^{Cre}$ and $Was^{fl/fl}Itgax^{Cre}$ mice. $Was^{fl/fl}LysM^{Cre}$ and $Was^{fl/fl}CD11c^{Cre}$ mice were subsequently crossed with $Rag1^{-/-}$ (The Jackson Laboratory, Stock No.002216) or $Rag2^{-/-}$ (Taconic, Stock No. RAGN12) mice to generate $Was^{fl/fl}LysM^{Cre}Rag1^{-/-}$ ($Was^{mDel}Rag1^{-/-}$) and $Was^{fl/fl}Itgax^{Cre}Rag2^{-/-}$ ($Was^{dcDel}Rag2^{-/-}$) mice, respectively, for CD4[+] T-cell-transfer-induced colitis studies. $Was^{-/-}$, $Was^{-/-}Rag2^{-/-}$ and $Rag2^{-/-}$ mice in 129SvEv background were generated as previously described[12]. $Il1r^{-/-}$ mice were obtained from Jackson Laboratories; Dr. Vijay Kuchroo at Brigham and Women Hospital, Boston, kindly provided $Il23r^{-/-}$ mice[53] and Dr. Raif Geha at Boston Children's provided $Dock8^{-/-}$ mice[36]. All the mice were housed in micro-isolator cages in a specific pathogen-free animal facility at Boston Children's Hospital. Sex- and age-matched animals between 5 and 14 weeks of age were used for experiments. We did not use randomization to assign animals to experimental groups. Investigators were not blinded to group allocation during experiments. No animals were excluded from the analysis. Mice were euthanasia by exposure to $CO_2$. Experiments were conducted after approval from the Animal Resources at Children's Hospital and according to regulations of the Institutional Animal Care and Use Committees (IACUC).

**Isolation of LP cells**. Cells were isolated from LP as described previously[30]. Briefly, colons were removed and placed in ice-cold phosphate-buffered saline (PBS) and the intestine is cut open longitudinally. Roughly 1.5 cm pieces of colon tissues were incubated in Hank's balanced salt solution (HBSS) containing 2% fetal bovine serum (FBS), 10 mM EDTA, 1 mM dithiothreitol and 5 mM HEPES at 37 °C with

shaking for 30 min to remove the epithelial cell layer. After removal of the epithelial layer, tissues were washed in PBS and digested in buffer containing HBSS, 20% FBS and collagenase VIII (200 unit/ml) for 60 min. The cells from digested tissues were filtered and washed once in cold PBS before re-suspending in 40% Percoll. Cells suspension was overlaid on 80% Percoll and centrifuged for 20 min at 2000 r.p.m. at room temperature. LP cells were collected from the interface of the Percoll gradient. LP macrophages were gated as $CD45^+CD11b^+CD64^+CD103^-$ cells as described by Bain et al.[54]. Finally, different macrophage populations were distinguished based on the expression of Ly6c and MHC II (Supplementary Fig. 1a).

**Generation of bone marrow chimera mice.** CD45.2 $Was^{-/-}$ recipient mice were irradiated with 1000 rad (Gamma Cell 40, $^{137}$Cs) in 2 doses of 500 rad each 4 h apart. Bone marrow cells from both femurs and tibiae of B6.SJL (CD45.1) and $Was^{-/-}$ (CD45.2) donor mice were collected under sterile conditions and suspended in PBS for injection. Bone marrow cells from B6.SJL (CD45.1) and $Was^{-/-}$ (CD45.2) mice were mixed at a 1:1 ratio and injected intravenously into each recipient mouse. The ratio of the bone marrow cells was confirmed by flow cytometry. Recipient mice were housed under specific pathogen-free conditions and were provided autoclaved water with sulfatrim (trimethoprim-sulfamethoxazole) and fed autoclaved food for 3 weeks. After 3 weeks, they were provided normal food and water. Eight to 10 weeks after the injection macrophage population in the colon were examined.

**Induction of colitis.** In this study colitis was induced by transferring naïve $CD45RB^{hi}CD4^+$T cell in 8- to 10-week-old $Rag1^{-/-}$, $Rag2^{-/-}$, $Was^{mDel}Rag1^{-/-}$ and $Was^{dcDel}Rag2^{-/-}$ mice. Splenocytes and lymph node cells form indicated mouse were enriched for $CD4^+$ T cells by magnetic-activated cell sorting (MACS) using mouse $CD4^+$ T cells isolation kit II (Miltenyi Biotec, Catalog Number 130-104-454). Naïve $CD4^+$ T cells ($TCR\beta^+CD4^+CD45RB^{high}$) were fluorescence-activated cell sorting (FACS) sorted from MACS-enriched $CD4^+$ T cells. Purity after sorting was > 98%. Naïve $CD4^+$ T cells ($3–5 \times 10^5$) was transferred intraperitoneally (i.p.) into each recipient. Body weight was monitored on a weekly basis.

In the macrophage-mediated colitis rescue experiments $1 \times 10^6$ unfractionated $CD4^+$ T cells was transferred i.p. into $Was^{-/-}Rag2^{-/-}$ and $Rag2^{-/-}$ mice. One day before T-cell transfer, mouse were treated with $2 \times 10^6$ macrophages form WT or $Was^{-/-}$ mice. Body weight was monitored on a weekly basis.

**Imaging and histology scoring.** An upright microscope (BX-41; Olympus) with bright-field and epifluorescence capability with SPOT imaging software and a DP70 color charge-coupled device camera were used for imaging colonic tissue A blinded reviewer independently scored haematoxylin and eosin stained colonic sections. Score range from 0–9 divided into three categories. Inflammation: 0, no inflammatory cells; 1, increased LP inflammatory cells; 2, confluence of inflammatory cells; 3, transmural extension of infiltrate. Crypt abscesses: 0, no abscess; 1, one to two abscess; 2, two to four abscess; 3, more then 4 abscess. Hyperplasia: 0, normal epithelium; 1, two time normal epithelial thickness; 2, more then two time normal thickness and reduction in Goblet cells; 3, three time normal thickness and few or no goblet cells.

**BMDM generation and polarization.** Bone marrow cells from femurs and tibiae were collected by flushing the bone using 25 G needle. After red blood cell lyses, cells were cultured in untreated 100 mm petri dish in Dulbecco's modified Eagle's medium (DMEM) containing 20% heat-inactivated fetal calf serum (FCS), penicillin–streptomycin and 30% L-cell conditioned medium. L-cell conditioned medium comprise of supernatant from culture of L929 cells that secretes macrophage colony-stimulating factor (M-CSF). Cells were fed with fresh medium every 3 days and finally collected at day 6. Collected cells were washed and cultured in the presence of combination of cytokine to polarize them into M0, M1 or M2r type of macrophages for 24 h. M0, culture in DMEM with 10% FCS; M1, cultured in the presence of LPS (100 ng/ml) and IFN-γ (20ng/ml); and M2r, cultured in the presence of IL-4 (20 ng/ml), IL-10 (20 ng/ml) and TGF-β (20 ng/ml)[28,29]. In some experiments, M2r macrophages were restimulated with LPS (10 ng/ml).

**Adoptive transfer of macrophages.** Bone marrow macrophages were stimulated with IL-4/IL-10/TGF-β for 24 h and each mouse was injected with two million macrophage intraperitoneally with PBS as vehicle. In few experiments, macrophage were labelled for 5 min at 37 °C with 3.5 mg/ml of DiR (1,19-dioctadecyl-3,3,39,39-tetramethylindotricarbocyanine iodide) before injecting them. Seven days post injection, cells were isolated from MLN and LP, and analysed in FACS for the presence of labelled macrophages.

**In vitro T-cell proliferation and Treg generation.** T-cell proliferation in the presence of different type of macrophage was determined by CFSE (carboxy-fluorescein succinimidyl ester) dilution. Naïve $CD4^+$ T ($CD4^+CD25^-CD44^-$) cells from splenocytes were isolated by MACS using mouse naïve T cells isolation kit (Miltenyi Biotec). Isolated $CD4^+$ T cells were stained with 5 mM CFSE for 5 min and washed extensively in PBS containing FCS. CFSE labelling of the responder

T cells were confirmed by flow cytometry. CFSE-labelled T cells ($1 \times 10^5$) were cultured with polarized macrophages at the ration of 4:1 (T cell:macrophage) in the presence of soluble anti-CD3ε (2 μg/ml)(eBioscience) for 4 days. Percentage of CFSE dilution was determined by flow cytometric analysis.

For in vitro Treg generation, we co-cultured MACS-sorted WT naïve $CD4^+$ T ($CD4^+CD25^-CD44^-$) with polarized macrophages at the ration of 4:1 (T cell:macrophage) in the presence of soluble anti-CD3e (2 μg/ml) and TGF-β (2 ng/ml). MACS sorted naïve T cells has < 1% FoxP3$^+$ cells. The percentage of Treg generation was determined after 5 days of co-culture by flow cytometry.

**Generation of human macrophage.** Human monocytes were isolated from healthy donors and patients with WAS in two-step process. First PBMCs were isolated from the buffy coat by Ficoll gradient centrifugation followed by MACS isolation of monocytes using Monocyte Isolation kit II (Miltenyi Biotec) as previously described[30]. Macrophages were obtained by culturing monocytes for 7 days in RPMI 1640 (Invitrogen) supplemented with 20% FCS (Invitrogen) and 100 ng/ml M-CSF (Miltenyi Biotec). Macrophage polarization was obtained by removing the culture medium and culturing cells for an additional 24 h in RPMI 1640 supplemented with 100 ng/ml LPS plus 20 ng/ml IFN-γ for M1 polarization or IL-4 (20 ng/ml) plus IL-10 (20 ng/ml), plus TGF-β (20 ng/ml), for M2r polarization[55]. Polarized macrophages were either analysed for gene expression or were co-cultured with naïve T cells to study in vitro Treg generation and T-cell proliferation. For in vitro Treg generation we co-cultured MACS-sorted $CD4^+CD25^-$ T cells with polarized macrophages at the ration of 4:1 (T cell:macrophage) in the presence of soluble anti-CD3e (2 μg/ml) and TGF-β (2 ng/ml). The percentage of Treg generation was determined after 5 days of co-culture by flow cytometry.

**Quantitative RT-PCR.** RNA from colonic tissue and BMDM was isolated using TRIzol reagent (Life Technologies) and complementary DNA was reverse transcribed from 1 μg total RNA using iScript cDNA Synthesis Kit (Bio-Rad). mRNA expression was measured using iQ SYBR Green on a CFX96 Real-Time System (Bio-Rad). In other experiments, different colonic macrophage population were FACS sorted into buffer RLT (Qiagen) and mRNA was isolated using RNeasy Plus mini kit (Qiagen). Expression of the transcripts was normalized against hypoxanthine-guanine phosphoribosyltransferase and presented as either relative expression ($2^{-dCt}$) or normalized fold change was calculated using the ΔΔCt method against mean control ΔCt (untreated WT for WT and $Was^{-/-}$ M1 or M2 macrophage in BMDM experiments, or macrophage from a healthy paired subject in experiments using monocyte-derived macrophage from $WASP$-deficient patients).

**Enzyme-linked immunosorbent assay.** Colonic tissue samples from mice were snap frozen and stored in − 80 °C for cytokine analysis. Frozen samples were homogenized in ice-cold PBS containing 1% NP-40 and protease inhibitor cocktail (Roche). Equal amount of protein is used to measure cytokines expression by enzyme-linked immunosorbent assay using commercially available kits (Biolegend).

**Flow cytometry.** Colonic LP cells were isolated as previously described. LP cells ($3–5 \times 10^6$) were resuspended in 100 μl flow cytometric staining buffer (2% FBS plus 0.1% NaN3 in PBS). Fc receptors were blocked for 10 min using anti-CD16/32 antibody (Biolegend) and surface antigens were stained for 30 min at 4 °C and washed with flow cytometric staining buffer. Zombie Violet Fixable Viability Stain (Life Technologies, Thermo Fisher Scientific) was used to eliminate dead cells, and forward- and side-scatter parameters were used for exclusion of doublets from analysis. Cellular fluorescence was assessed using a BD Canto II Flow Cytometer (BD Biosciences), and percentages of subsets and mean fluorescence intensity were analysed with FlowJo software, versions 9 and 10.0.8. For intracellular FOXP3 staining, cells were fixed and permeabilized with the Fixation/Permeabilization Solution Kit (eBioscience) according to the manufacturer's instructions after surface staining. Cells were stained for 30 min at room temperature with Abs and washed twice with the permeabilization buffer. For intracellular staining of cytokines (TNF, IL-17A, IL-10, and IFN-γ), cells were incubated with PMA (10 ng/ml), ionomycin (500 ng/ ml) and GolgiStop (BD Biosciences) in RPMI Media 1640 (Life Technologies) supplemented with 10% FBS and antibiotics for 5 h at 37 °C, to stimulate cytokine production. Fluorescence-labelled Abs used were listed in Supplementary Table 1.

**Western blotting and immunoprecipitation.** After treating the BMDM as mentioned cells were lysed in RIPA buffer on ice for 1 h. For the immunoblot analysis, 30 μg of protein were resolved by 4–20% gradient SDS-polyacrylamide gel electrophoresis (PAGE) and transferred to nitrocellulose membranes. The membranes were blocked with 5% non-fat dry milk in TBS-T (0.1% Tween 20) for 1 h before incubation overnight at 4 °C with primary antibodies. The membranes were then washed and incubated with horseradish peroxidase-conjugated secondary antibodies in 5% non-fat dry milk in TBS-T for 1 h. After successive washes, the membranes were developed using a SuperSignal West Pico Chemiluminescent kit (Thermo Scientific). Immunoprecipitations with anti-WASP antibodies were performed on pre-cleared BMDM cell lysate using Protein A/G magnetic beads

(Thermo Scientific) at 4 °C for 14–16 h. The beads were washed three times in washing buffer (20 mM Tris-HCl pH 7.4 and 0.1% Nonidet P-40) and samples were boiled for 10 min in 25 μl of loading buffer and subjected to SDS-PAGE and immunoblot analysis. Antibodies used were listed in Supplementary Table 1.

**Statistics**. All data were analysed by Student's *t*-test with 95% confidence interval or analysis of variance using GraphPad Prism version 6.0 (GraphPad Software) and presented as mean ± SEM. Normal distribution was assumed. A *p*-value of < 0.05 was considered statistically significant. (*$P < 0.05$, **$P < 0.01$, ***$P < 0.001$ and ****$P < 0.0001$; NS, not significant).

**Study approval**. All patients provided written informed consent prior to inclusion in the study. Clinical patient samples were collected under a Boston Children's Hospital IRB-approved research protocol (P00000529). All animal experiments were performed in accordance with Institutional Animal Care and Use Committee-approved protocols number 14-04-2677 R (IACUC, Boston Children's Hospital) and adhered to the National Research Council's 'Guide to the care and Use of Laboratory Animals'.

**Data availability**. The authors declare that the data supporting the findings of this study are available within the article and its Supplementary Information files, or are available upon reasonable requests to the authors.

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

## Acknowledgements

A.B. is recipient of CCFA Career Development Award (327200) and NIH KO1 award (K01DK109026). S.B.S is supported by NIH Grants HL59561, DK034854 and AI50950, the Helmsley Charitable Trust and the Wolpow Family Chair in IBD Treatment and Research. LDN is supported by Division of Intramural Research, NIAID, NIH.

## Author contributions

Conceived and designed the experiments: A.B. and S.B.S. Performed experiments: A.B., D.S., A.G., M.F., J.A.G., L.K., Y.H.K. and N.S.R. Acquired and analysed data: A.B. and D.S. Provided reagents and clinical patient samples: E.J., R.S.G., A.T., V.K.K., L.D.N. and S.Y.P. Wrote the manuscript: A.B. and S.B.S. Edited the manuscript: D.S. and B.H.H. Critical review of data and provided suggestions: T.C. and B.H.H. All authors approved the final version of the manuscript.

## Additional information

**Competing interests:** The authors declare no competing interests.

