## [Peer Review File · Nature Communications]

Reviewers' comments:

Reviewer #1 (Colitis, transcriptional regulation)(Remarks to the Author):

This study builds on previous work from the Snapper lab on the function of the WASP gene in intestinal homeostasis. The new results show that WASP mediates immune regulatory functions in intestinal macrophages promoting a more anti-inflammatory phenotype

Deficiency in WASP is shown to make colonic macrophages more pro-inflammatory, with IL-1 and IL-23 being amongst the macrophage products that drive the pathological changes in the gut of *was*^{-/-} mice. Using a mixed bone marrow chimera approach the authors demonstrate that WASP deficiency acts on gut macrophages in a cell intrinsic way. Further in vitro investigation shows that IL-10-induced STAT3 phosphorylation is diminished in bone marrow derived WASP^{-/-} macrophages. The authors present evidence that the WASP-Dock8 signalling complex is required for optimal signalling of the IL-10/ Stat3 axis and that Dock8 deficiency phenocopies the absence of WASP in this in vitro system.

Overall this study is presented in a well-organised way and informs our understanding of how WASP and IL-10 function in the gut at a molecular level. The evidence linking a WASP/Dock 8 interaction to IL-10 signaling and immune regulatory macrophage function is the weakest and is preliminary as presented. The data are over-interpreted particularly in the Title and Abstract of the ms.

Specific Comments:

1. Throughout the ms data on frequencies and total numbers of cells should be presented. Some Figs contain only one measure.
2. Information on the number of times experiments have been repeated should be included for all Figs. This is mentioned in some but not other legends
3. Gene expression as shown in Fig 1 c should be shown in the in bm chimera experiments (Fig 1d) to show similar cell intrinsic skew in inflammatory cytokine production in the *was*^{-/-} macrophages
4. There should be further characterisation of key myeloid progenitor populations in *was*^{-/-} mice to ensure there are not deficiencies in progenitors that are responsible for the observed phenotypes. Related to this, the phenotype of starting and end populations in BMDM cultures of *was*^{-/-} versus control bmdm cultures should be shown. Are there differences in IL-10 or TGF-beta receptor expression in *was*^{-/-} mice that could account for the differences?
5. Fig 2 T cell transfer should be compared in whole *was* ko as well as cell type specific ones to show knock out in one population recapitulates the phenotype of the whole RAG^{-/-} mouse.
6. Fig 5 what is happening to the transferred macrophages, are there differences in homing of KO and WT macrophages to the intestine that explain the different outcomes in intestinal inflammation.
7. Fig 6 The objective of these studies is to show that the phenotype of WASP KO macrophages is due to impaired IL-10 signalling which involves a WASP/DOCK 8 complex. This is all based on immunoprecipitation and the finding that WASP^{-/-} and DOCK 8^{-/-} BMDM cells phenocopy each other. It is possible that WASP and DOCK 8 mediate these similar phenotypes through distinct mechanisms especially as WASP is known to mediate a range of functions in macrophages. Evidence to support a requirement for the WASP/DOCK 8 complex in the inflammatory phenotype of M2R cultured BMDM would help as would reversal of phenotype by overexpression of STAT 3. In Fig 6 F it would be helpful to know if the phenotype in WASP^{-/-} or Dock8^{-/-} BMDM is dependent on addition of IL-10 in the differentiation process. What about expression of IL-10R and TGF-BR1? Dose of IL-10 should be indicated.

Reviewer #2 (Myelopoiesis, macrophage)(Remarks to the Author):

Snapper and colleagues describe the effects of WASp mutations (and Dock8) on several different immunoregulatory pathways in myeloid cells. They show (1) that myeloid to macrophage development (in the lamina propria) is partially arrested, that (2) there are defects in M2 polarization, and (3) there are effects in the IL-10 pathway. Potentially, any of these WASp-dependent pathways may be of interest in understanding myeloid cell biology. However, the main problem with the manuscript is the mechanistic links between the pathways. In some ways, it seems that the manuscript tried to tie together a few stories. Some specific comments are:

Major:

1. In the first part of the results, the authors make a big deal about LP macrophage development. However, the 'P1 – P4' transition is a canonical pathway for all monocytes developing into macrophages (skin, tumors, etc) as shown in many papers such as Bain et al.. The authors show WASp-deficient cells have a cell autonomous defect (partial) in the normal developmental transition, but don't have a convincing explanation of why this occurs. Can this phenotype be recapitulated in vitro (for example, from a GM-CSF culture, which better reflects the Ly6C>MHCII transition compared to a CSF-1 culture?)
2. Following on from point 1, it is also unsurprising that there is relatively less M2 markers – there are less mature macrophages.
3. In figure 4, the authors did not seem to consider the fact that the increased TNF can block the M2 pathway as shown in mice lacking TNF or the TNFR1 (work by Murray, Koener, Bogdan). If WASp-deficient mice are treated with anti-TNF (or macrophages in vitro) is the M2 defect reversed? The link to TNF is a critical point that must be addressed, as the authors are experts in human IBD, a disease often driven by TNF.
4. In Figure 4c, the 'defect' is probably simply due to the reduced amounts of Arg1 – can the defect be rescued by exogenous arginine (see Van de Velde et al. JBC for example)?
5. The data in Figure 4 is correlative with the rest of the manuscript: there is no direct evidence the "M2r" macrophages (whatever they are) are causative of the phenotypes.
6. Figure 6 is confusing. The authors state there is nothing wrong with Stat6p which does not seem to be the case from the actual data (e.g. 60 minutes). This data should be buttressed with phospho-flow data. The data from Stat3p seems convincing, but again there is not direct evidence of what is going on. What happens to surface IL-4Ra amounts? Since IL-4ra is a direct IL-10-Stat3 target, does this account for the M2 defect (see Lang et al, 2002, Figure 6). None of the canonical IL-10 target genes were checked.

Other comments:

1. The first paragraph of the introduction is boring.

In summary, while many aspects of this study are interesting, the cause-versus-consequence for the data remains a limitation that could be addressed.

Reviewers' comments:

Reviewer #1:

1. Throughout the manuscript data on frequencies and total numbers of cells should be presented. Some Figs contain only one measure.

In the revised manuscript, we have included both frequency and absolute cell numbers in **Figures 1, 2, 3 and 5**. In Figure 1c, we did not present the absolute number of macrophage as the reconstitution of $Was^{-/-}$ and wild type cells are not similar in the LP. Cells in wild type (CD45.1) compartment are twice as much as from the $Was^{-/-}$ (CD45.2) compartment. We have previously shown that expression of WASP confers a selective advantage to hematopoietic cell (Westerberg et al. Blood 2018). Therefore, for this scenario, the absolute number data will not be informative. Nonetheless, we present the data below (**Figure R1**):

Figure R1: CD45.1⁺ (WT) and CD45.2⁺ ($Was^{-/-}$) bone marrow cells were transferred at the ratio of 1:1 into lethally irradiated CD45.2⁺ $Was^{-/-}$ recipient. LP M ϕ was analyzed after 10 weeks. Graph shows the quantification of P2 and P3/P4 cells in the WT (n=6) and $Was^{-/-}$ (n=6) compartment of chimeric mice.

2. Information on the number of times experiments have been repeated should be included for all Figs. This is mentioned in some but not other legends.

In the revised manuscript, we have mentioned in the figure legends the number of times each experiment was conducted.

3. Gene expression as shown in Fig 1c should be shown in the bone marrow chimera experiments (Fig 1d) to show similar cell intrinsic skewing in inflammatory cytokine production in the $Was^{-/-}$ macrophages

As suggested by the reviewer, we sorted the P3/P4 macrophages from the bone marrow chimera mice and analyzed the expression of pro- and anti-inflammatory genes in the wild type (CD45.1) and $Was^{-/-}$ (CD45.2) compartment. Data are presented as **Figure 1e** in the revised manuscript.

4. There should be further characterization of key myeloid progenitor populations in $Was^{-/-}$ mice to ensure there are not deficiencies in progenitors that are responsible for the observed phenotypes.

As suggested by the reviewer, we analyzed the myeloid progenitor in the bone marrow of wild type and $Was^{-/-}$ mice. The frequency of granulocyte/macrophage progenitor (GMPs: Lin⁻CD127⁻cKit⁺Sca1⁻CD16/32⁺CD34⁺) and common myeloid progenitors (CMPs: Lin⁻CD127⁻cKit⁺Sca1⁻CD16/32⁻CD34⁺) were comparable between wild type and $Was^{-/-}$ mice. Data are presented below (**Figure R2a**). A comment related to the lack of differences in myeloid progenitors between wild type and $Was^{-/-}$ mice is added as data not shown in the revised manuscript.

5. Related to this, the phenotype of starting and end populations in BMDM cultures of $Was^{-/-}$ versus control BMDM cultures should be shown.

To examine if there is any difference in the starting and end population of BMDM culture between wild type and *Was*^{-/-} mice, we analyzed the expression of M2 specific markers (*Arg1*, *Fizz1* and *Ym1*) in the bone marrow cells (starting population) and the M0 macrophages (end population of BMDM culture). We did not observe any significant differences in the expression of *Arg1*, *Fizz1* and *Ym1* between the wild type and *Was*^{-/-} mice, either in bone marrow or M0 macrophages (**Figure R2b&c**). These data suggest that the starting and end population of BMDM culture is comparable between wild type and *Was*^{-/-} mice. A comment related to the lack of differences of M2 specific markers in the bone marrow

(i.e., progenitors as discussed in comment above) or M0 macrophage populations between wild type and *Was*^{-/-} mice is added as data not shown in the revised manuscript.

4c. Are there differences in IL-10 or TGF-beta receptor expression in *Was*^{-/-} mice that could account for the differences?

The reviewer raises a valid point. We agree with this concern, and to answer that we analyzed the expression of IL10R α , IL10R β , TGF β 1, TGF β 2 and TGF β 3 in the BMDM from wild type and *Was*^{-/-} mice by qPCR. We were unable to examine the surface expression of the receptors by flow cytometry due to lack of reliable antibodies. At steady state we did not find any difference in the expression of IL10 and TGF receptor transcript between wild type and *Was*^{-/-} mice. The data are discussed in the Results section and presented in the revised manuscript as supplemental **Figure 6a**.

6. Fig 2 T cell transfer should be compared in whole *Was*^{-/-} as well as cell type specific ones to show knock out in one population recapitulates the phenotype of the whole *RAG*^{-/-} mouse.

As suggested by the reviewer, we have now incorporated the T cell transfer induced colitis data from the *Was*^{-/-Rag1}^{-/-} mice in the **Figure 2a-b**. We observed similar body weight loss and histological

Figure R2: (a) Flow cytometric analysis of bone marrow cells from wild type and *Was*^{-/-} mice. (WT n=3; *Was*^{-/-} n=3) **(b)** qPCR analysis of M2 specific gene expression (WT n=3; *Was*^{-/-} n=3) in bone marrow cells. **(c)** Bone marrow cells were differentiated into macrophages in presence of MCSF for 7 days. Expression of M2 specific genes was analyzed by qPCR (WT n=3; *Was*^{-/-} n=3).

disease in *Was^{-/-}Rag1^{-/-}* and *Was^{mDel}Rag1^{-/-}* mice.

7. Fig 5 what is happening to the transferred macrophages, are their differences in homing of KO and WT macrophages to the intestine that explain the different outcomes in intestinal inflammation.

In order to identify whether transferred macrophages migrate to the LP, macrophages were labeled with XenoLight DiR, (1,1'-dioctadecyltetramethyl indotricarbocyanine Iodide) (Invitrogen, CA) prior to IP injection. After 7 days presence of labeled macrophages in the LP was determined by flow cytometry. Data is presented in **Supplemental Figure 5a**. We observed similar frequencies of wild type and *Was^{-/-}* M2r macrophages in the lamina propria.

8. Fig 6 the objective of these studies is to show that the phenotype of WASP KO macrophages is due to impaired IL-10 signaling which involves a WASP/DOCK 8 complex. This is all based on immunoprecipitation and the finding that *Was^{-/-}* and *Dock8^{-/-}* BMDM cells phenocopy each other. It is possible that WASP and DOCK8 mediate these similar phenotypes through distinct mechanisms especially as WASP is known to mediate a range of functions in macrophages. Evidence to support a requirement for the WASP/DOCK8 complex in the inflammatory phenotype of M2R cultured BMDM would help as would reversal of phenotype by overexpression of STAT3.

We thank the reviewer for this comment. As suggested we have now provided the data showing the involvement of WASP/DOCK8 complex in M2r macrophages. We observed enrichment of WASP/DOCK8 complex in M2r conditioned macrophages by co-immunoprecipitation (**Figure 6f**).

We also performed the STAT3 overexpression experiment. Wild type and *Was^{-/-}* BMDM were transduced by retroviral transduction of empty vector (pMSCV-IRES-Thy1.1) or HA-tagged murine STAT3 (pMSCV-HA-STAT3-IRES-Thy1.1) (**Figure R3a**).

[Cloning strategy: the STAT3 construct was cloned from a pcDNA3-HA-STAT3 vector (a gift from Dr Bruce Horwitz) into a pMSCV-IRES-Thy1.1 transfer vector. The pMSCV-IRES-Thy1.1 vector was generated by replacing the GFP from a pMSCV-IRES-GFP II vector (a gift from Dr. Dario Vignali, Addgene plasmid # 52107) with a Thy1.1 construct. The Thy1.1 construct was cloned from a pcDNA3-Thy1.1 plasmid (a gift from Dr Wayne Lencer). pCL-Eco packaging plasmid was a gift from Dr. Jon Kagan.]

We used the Thy1.1 surface expression to select and enrich transduced macrophages using MACS sorting. Transduction in wild type and *Was^{-/-}* BMDM were comparable (**Figure R3b**). We observed an increase in total STAT3 in both wild type and *Was^{-/-}* BMDM transduced with HA-STAT3 vector compared to empty vector (**Figure R3c**). Thereafter, transduced macrophages were either stimulated in presence of IL10 or M2r-polarized in presence of IL10, IL4 and TGF β . STAT3 phosphorylation was analyzed using flow cytometry. As expected, IL10 induced higher STAT3 phosphorylation in wild type macrophages compared to *Was^{-/-}* macrophages transduced with empty vector. Over expression of STAT3 lead to increase in phosphor-STAT3 in both wild type and *Was^{-/-}* macrophages in presence of

IL10. However, the STAT3 phosphorylation in HA-STAT3 transduced *Was^{-/-}* macrophage was still less than HA-STAT3 transduced wild-type macrophages (**Figure R3d**). Similarly, expression of M2-markers (Arg1, Ym1 and Fizz1) was increased following STAT3 overexpression in both wild type and *Was^{-/-}* M2r macrophages. However, their expression was significantly less in *Was^{-/-}* compared to wild type macrophages in both empty vector and HA-STAT3 transduced M2r macrophages. Interestingly, we found that overexpression of STAT3 in *Was^{-/-}* macrophages increased the level of M2 specific genes compared to empty vector (**Figure R3e**). Taken together we can conclude that STAT3 overexpression in *Was^{-/-}* macrophages partially rescues the M2r phenotype but does not reach the level of wild type

(SV). These data signify that lack of WASP affects IL10 mediated STAT3 phosphorylation even when the total STAT3 is in abundance.

Figure R3: Overexpression of STAT3 partially rescues the M2r polarization in Was^{-/-} macrophages. a) Schematic of empty vector (EV) and HA-STAT3 vector (SV) **b)** Retroviral transduced macrophages were sorted on MACS using CD90.1 microbeads (Miltenyi Biotec), and expression of Thy1.1 was analyzed by flow cytometry. Numbers indicated percent of cells positive for Thy1.1 expression **c)** Expression of total STAT3 was analyzed in viral transduced macrophages. **d)** Macrophages were stimulated in presence of IL10 for indicated time and pSTAT3 expression was analyzed by flow cytometry. MFI: mean fluorescence intensity **e)** Viral transduced macrophages were polarized to M2r macrophages in presence of IL10 (20ng/ml), IL4 (20ng/ml) and TGFβ (20ng/ml). Expression of M2-specific markers (*Fizz1* and *Arg1*) was examined by qPCR using specific primers. . *p < 0.05, **p < 0.01, ***p < 0.001 (Unpaired Student's t-test).

9. In Fig 6 F it would be helpful to know if the phenotype in *Was*^{-/-} or *Dock8*^{-/-} BMDM is dependent on addition of IL-10 in the differentiation process. What about expression of IL-10R and TGF-BR1? Dose of IL-10 should be indicated.

We performed macrophage polarization in presence of IL10, IL4 and TGFβ (M2r conditioning) or IL4 and TGFβ, and analyzed the expression of M2-specific markers. There was markedly reduced expression of the M2 markers, *Arg1*, *Ym1* and *Fizz1*, in IL4 and TGFβ treated macrophages compared to IL10, IL4 and TGFβ treated M2r macrophages both in wild type and *Was*^{-/-} macrophages (**Supplemental Figure 6c**). As shown in Figure 4, we observed a similar reduction in expression of all M2 markers in *Was*^{-/-} compared to wild type M2r macrophages. However, the expression of M2 markers was not significantly different between *Was*^{-/-} and wild type macrophages when conditioned in presence of IL4 and TGFβ. Among *Arg1*, *Ym1* and *Fizz1*, the expression of only *Arg1* showed statistically significant reduction in absence of WASP (**Supplemental Figure 6c**). Therefore, from these observations, we can conclude that the defect in the M2r polarization of *Was*^{-/-} macrophages is largely contributed by the aberrant IL10 signaling in absence of WASP.

At steady state, we did not find any difference in the expression of IL10 and TGF receptor transcript between wild type and *Was*^{-/-} mice. The data are discussed in the Result section and presented in the revised manuscript as **Supplemental Figure 6a**.

In the revised manuscript, we mentioned the dose of IL10 in the figure legends.

Reviewer #2:

1. In the first part of the results, the authors make a big deal about LP macrophage development. However, the 'P1–P4' transition is a canonical pathway for all monocytes developing into macrophages (skin, tumors, etc) as shown in many papers such as Bain et al. The authors show WASP-deficient cells have a cell autonomous defect (partial) in the normal developmental transition, but don't have a convincing explanation of why this occurs. Can this phenotype be recapitulated *in vitro* (for example, from a GM-CSF culture, which better reflects the Ly6C⁺MHCII⁺ transition compared to a CSF-1 culture)?

We agree with the reviewer that the P1 to P4 transition is a canonical pathway of tissue macrophage development from monocyte. However, we have previously published that the defect in IL10 signaling leads to aberrant monocyte to macrophage transition in intestinal LP (*Shouval et al. Immunity 2014; Redhu et al. eLife 2017*). In this manuscript, we show that the P1 to P4 transition of LM macrophages is defective in the absence of WASP. Furthermore we found that the *in vitro* generation and function of anti-inflammatory M2r macrophages from BMDM is aberrant in *Was*^{-/-} mice and macrophage-specific deletion of *Was*^{-/-} lead to exacerbated T cell transfer induced colitis. Finally, we show that in absence of WASP, IL10 signaling is defective. Collectively, our published and presented data strongly suggest that the macrophage polarization defect that we observe in absence of WASP both *in vitro* and *in vivo* is probably due to defect in IL10 signaling.

As suggested by the reviewer we tried to recapitulate the *in vivo* LP macrophage transition in an *in vitro* BMDM culture using GMCSF and MCSF (**Figure R4a-b**). Unfortunately, we were unable to visualize the P1 to P4 transition *in vitro* as observed in the intestinal LP using the gating strategy described in supplemental Figure 1a. To our knowledge, in contrast to humans, there have been no reports with murine BMDM a strategy *in vitro* to differentiate intestinal like macrophages. Differentiation in presence of GMCSF leads to generation of mostly Ly6c⁺MHCII⁻ monocyte-type cells.

Figure R4. Flow cytometric analysis of BMDM differentiated in presence of (a) MCSF and (b) GMCSF. Macrophages were gated as live CD45⁺CD11b⁺CD103⁻CD64⁺ cells (WT n=3; Was^{-/-} n=3).

2. Following on from point 1, it is also unsurprising that there are relatively less M2 markers – there are less mature macrophages.

In Figure 1b we analyzed the expression of pro- and anti-inflammatory genes in FACS sorted LP macrophages; the gene expression data was presented after normalization against a house-keeping gene (HPRT). We understand the reviewer's concern; however, we think the difference in the expression of pro- and anti-inflammatory genes is not due the difference in percentage of P3/P4 macrophages between wild type and Was^{-/-} mice. In the revised Figure 1a we have also presented the absolute number of macrophages in the LP of wild type and Was^{-/-} mice, which show that the total number of P3/P4 macrophages is increased in the Was^{-/-} compared to wild type mice. Moreover, if the reduced expression of anti-inflammatory gene in Was^{-/-} mice were due to less frequency of P3/P4 cells, it would have also led to reduced expression of inflammatory genes; however, we observed that there is actually an increase in the expression of inflammatory genes. Therefore, taken together we believe that the P3/P4 macrophages in Was^{-/-} mice are functionally more inflammatory compared to wild type P3/P4 macrophages.

3. In figure 4, the authors did not seem to consider the fact that the increased TNF can block the M2 pathway as shown in mice lacking TNF or the TNFR1 (work by Murray, Koener, Bogdan). If WASp-deficient mice are treated with anti-TNF (or macrophages in vitro) is the M2 defect reversed? The link to TNF is a critical point that must be addressed, as the authors are experts in human IBD, a disease often driven by TNF.

The reviewer raises a valid point: anti-TNF therapy has been shown to be effective in various IBD models including DSS-induced colitis and also in IBD patients. Moreover, TNF also blocks M2 polarization of macrophages. To address the reviewer's concern, we treated 12-week-old Was^{-/-} mice with mouse anti-TNF or isotype control Ab (Bio X cell). We injected 200 ug Ab/mice once a week for two weeks and examined the LP macrophage populations. Anti-TNF treatment in Was^{-/-} mice was unable to reverse the *in vivo* macrophage defect. Similar to the data in Figure 1, we observed an increase in the frequency of P2 (pro-inflammatory) macrophages and concomitant decrease in P3/P4 (anti-inflammatory) macrophages in the LP of Was^{-/-} mice treated with isotype Ab compared to wild type

mice (**Figure R5a**). These data indicate that TNF blockade in *Was*^{-/-} mice is insufficient to protect them from colitis. Our observation is in line with a previous report, where it has been shown that anti-TNF treatment fails to protect macrophage-specific IL10R deficient mice from DSS induced colitis (Li et al. Mucosal Immunol. 2014).

We also used anti-TNF Ab in M2r conditioning culture. Similar to our *in vivo* findings, we did not notice any change in the expression of M2-specific markers including Arg1, Ym1 and Fizz1 in *Was*^{-/-} macrophages in presence of anti-TNF Ab (**Figure R5b**).

To confirm the functional capacity of the anti-TNF Ab used in the above experiments, we induced colitis with DSS in wild type mice and treated them with the same anti-TNF Ab. The mice that were given anti-TNF Ab lost less weight compared to the isotype treated mice (**Figure R5c**).

Figure R5: Anti-TNF treatment alone did not rescue the aberrant macrophage function in *Was*^{-/-} mice. **(a)** 12 week old *Was*^{-/-} mice were treated with isotype-control Ab (250µg/mice, Bio X Cell) or anti-TNF Ab (250µg/mice, Bio X Cell) once a week for two weeks. LP macrophages were analyzed by flow cytometry in wild type and treated *Was*^{-/-} mice (WT n=4; *Was*^{-/-} + isotype n=4; *Was*^{-/-} + anti-TNF n=4). **(b)** BMDM was polarized with IL10 (20ng/ml), IL4 (20ng/ml) and TGFβ (20 ng/ml) in presence of isotype-control Ab (50 µg/ml) or anti-TNF Ab (50 µg/ml). Expression of Arg1 and Ym1 in polarized macrophages examined by qPCR. **p < 0.01, ***p < 0.001 (Unpaired Student's t-test). **(c)** Colitis was induced in wild type mice with 3% DSS. Anti-TNF (250 µg/mouse,) or isotype Ab was administered *i.p.* on day 4 and 8. Mean ± SEM of percent initial body weight were plotted. *p < 0.05, ****p < 0.0001 (Two-way ANOVA).

All in all, we believe that the anti-TNF therapy alone is not capable of reversing the colonic inflammation in WASP-deficiency where IL10 signaling is defective, and associated with elevated level of IL1β and IL23.

4. In Figure 4c, the 'defect' is probably simply due to the reduced amounts of Arg1 – can the defect be rescued by exogenous arginine (see Van de Velde et al. JBC for example)?

We agree with the reviewer's comment that increased T cell proliferation observed in the presence of *Was*^{-/-} compared to wild type M2r macrophages could be due to reduced expression of Arg1 (Arginase) in macrophages lacking WASP. The reviewer suggested to test if exogenous arginine could rescue the defect. Since in *Was*^{-/-} macrophages there is reduction in *Arg1* gene expression, we reasoned that using exogenous Arginase 1 would be a better option. We examined if adding Arginase 1 (R&D systems) in the co-culture system could inhibit T cell proliferation induced by *Was*^{-/-} M2r macrophages. Unfortunately, the Arg1 enzyme we used was not effective, as it was unable to inhibit T cell proliferation in either a Mφ:T cell co-culture experiment or a control experiment with in vitro stimulated naïve T cells (with plate bound anti-CD3 and anti-CD28) (**Figure R6a-b**). Since the experiment failed to produce conclusive results, we reached out to the group who provided PEG-Arg1 for the study in the JBC article mentioned by the reviewer (Van de Velde et al. JBC. 2017), but unfortunately, we did not receive a reply. Since we consider the role of *Agr1* in T cell proliferation is a valid point, we have included this point in the Results section with reference to the above study.

Figure R6: Naïve CD4⁺ T cells were MACS sorted and CFSE labeled. Labeled T cells were either cultured with plate bound anti-CD3 and anti-CD28 Ab (a) or in presence of plate bound anti-CD3 and *Was*^{-/-} M2r Mφ (b) and in presence or absence of Arg1. After 3 days T cell proliferation was determined by flow cytometry. Numbers denote % proliferation.

5. The data in Figure 4 is correlative with the rest of the manuscript: there is no direct evidence that the "M2r" macrophages (whatever they are) are causative of the phenotypes.

The data presented in this manuscript show that in absence of WASP, anti-inflammatory macrophage function is disrupted due to defective IL10 signaling (**Figure 6** and new **Supplemental Figure 6c**). Moreover, macrophage-specific deletion of *Was*^{-/-} led to exacerbated T cell transfer induced colitis. We also observed a similar defect in anti-inflammatory macrophage generation and function in WAS patients. In Figure 5 we showed that transfer of wild type but not *Was*^{-/-} M2r macrophages partially protected *Was*^{-/-} *Rag*^{-/-} mice from T cell transfer induced colitis. This data may not be direct evidence, but it clearly shows that the restoration of anti-inflammatory macrophage function can lead to reversal of disease in *Was*^{-/-} mice.

6. Figure 6 is confusing. The authors state there is nothing wrong with *Stat6p*, which does not seem to be the case from the actual data (e.g. 60 minutes). This data should be buttressed with phospho-flow data.

We analyzed STAT6 phosphorylation in presence of IL4 by phospho-flow and incorporated the data in supplemental **Figure 6b**. We did not observe any difference in STAT6 phosphorylation between wild type and *Was*^{-/-} macrophages.

7. The data from *Stat3p* seems convincing, but again there is not direct evidence of what is going on. What happens to surface IL-4Ra amounts? Since *IL-4ra* is a direct IL-10-*Stat3* target, does this account

for the M2 defect (see Lang et al, 2002, Figure 6). None of the canonical IL-10 target genes were checked.

We agree with the reviewer that the regulation of IL4R expression by IL10 could be a mechanism involved in WASP mediated regulation of M2 polarization. We analyzed the expression of IL4R both by qPCR and flow cytometry. We found that the expression of IL4R was significantly less in *Was*^{-/-} M2r macrophages compared to wild type macrophages. IL10 mediated induction of IL4R was also aberrant in *Was*^{-/-} macrophages. However, we found that IL4R expression was comparable in untreated wild type and *Was*^{-/-} macrophages (**Supplemental Figure 6d-e**). We also examined the expression of another IL10 target gene, SOCS3. Expression of SOCS3 was significantly reduced in *Was*^{-/-} M2r and IL10 stimulated macrophages compared to wild type macrophages (**Supplemental Figure 6d**). These findings further strengthen our hypothesis that WASP is involved in the regulation of IL10-mediated STAT3 signaling, and aberrant induction of IL4R expression by IL10 in *Was*^{-/-} macrophages could also be responsible for defective M2 polarization. These data are now presented in **Supplemental Figure 6d-e** and described in the Results section.

Other comments:

1. The first paragraph of the introduction is boring.

We appreciate that the first paragraph of the introduction, as initially written, was a basic review of innate immune cells and not particularly intellectually appealing for sophisticated immunologists. However, given that the audience for Nature Communications is a broad readership, we felt that a brief introduction to innate immune cells was needed. We have slightly condensed and restructured the introduction which we hope will be satisfactory for the Reviewer.

In summary, while many aspects of this study are interesting, the cause-versus-consequence for the data remains a limitation that could be addressed.

We believe that our manuscript has been significantly improved after responding to the reviewers' critiques. In the revised manuscript and response to reviewers we have presented new data that strengthen our hypothesis that the aberrant IL10 signaling is the predominant driver of the macrophage defect observed in *Was*^{-/-} mice (**Supplemental Figure 6c, d and e**). Partial rescue of M2-phenotype in *Was*^{-/-} macrophages by over expression of STAT3 also support our premise. We believe that these additional experiments address the critical issues raised by the Reviewers, providing further mechanistic insight.

Reviewer #1 (Remarks to the Author):

In my opinion the authors have improved the manuscript with the inclusion of new data and the manuscript is now suitable for publication. The authors should discuss the limitation that they have not directly shown a WASP/Dock 8 interaction is crucial for IL-10R signalling and further experiments are required. This conclusion should be removed from the title.

The authors should also mention the frequency of CD45 cells from WAS^{-/-} versus WT donors in the BM in the spleen, MLN and gut to determine if the reduced macrophage finding is reflected in other immune cells and present in systemic compartments as well as the intestine. The authors must have this data

The ms needs editing for grammatical errors

Reviewer #2 (Remarks to the Author):

The authors have done a comprehensive revision of their manuscript. The major questions were addressed primarily through careful experimentation.

Reviewer #1 (Remarks to the Author):

In my opinion the authors have improved the manuscript with the inclusion of new data and the manuscript is now suitable for publication. The authors should discuss the limitation that they have not directly shown a WASP/Dock 8 interaction is crucial for IL-10R signalling and further experiments are required. This conclusion should be removed from the title.

Reply: We have modified our conclusions in the title, abstract, result and discussion section.

The authors should also mention the frequency of CD45 cells from *Was*^{-/-} versus WT donors in the BM in the spleen, MLN and gut to determine if the reduced macrophage finding is reflected in other immune cells and present in systemic compartments as well as the intestine. The authors must have this data

Reply: We previously described in detail the selective advantage of WASP-expressing cells in bone marrow, spleen and MLN (Westerberg et al. Blood 2008). In our chimera experiment we also observed a similar selective advantage of WASP-expressing cells in the lamina propria (Figure 1d) and blood (shown below).

The ms needs editing for grammatical errors

Reply: We proof read the manuscript for mistakes.

Reviewer #2 (Remarks to the Author):

The authors have done a comprehensive revision of their manuscript. The major questions were addressed primarily through careful experimentation.

Reply: We thank the reviewer for his/her constructive comments.